# Learning to Cooperate with Humans using Generative Agents

**Yancheng Liang, Daphne Chen, Abhishek Gupta, Simon S. Du\*, Natasha Jaques\***
University of Washington
`{yancheng, daphc, abhgupta, ssdu, nj}@cs.washington.edu`

## Abstract

Training agents that can coordinate zero-shot with humans is a key mission in multi-agent reinforcement learning (MARL). Current algorithms focus on training simulated human partner policies which are then used to train a Cooperator agent. The simulated human is produced either through behavior cloning over a dataset of human cooperation behavior, or by using MARL to create a population of simulated agents. However, these approaches often struggle to produce a Cooperator that can coordinate well with real humans, since the simulated humans fail to cover the diverse strategies and styles employed by people in the real world. We show *learning a generative model of human partners* can effectively address this issue. Our model learns a latent variable representation of the human that can be regarded as encoding the human's unique strategy, intention, experience, or style. This generative model can be flexibly trained from any (human or neural policy) agent interaction data. By sampling from the latent space, we can use the generative model to produce different partners to train Cooperator agents. We evaluate our method—**G**enerative **A**gent **M**odeling for **M**ulti-agent **A**daptation (**GAMMA**)—on Overcooked, a challenging cooperative cooking game that has become a standard benchmark for zero-shot coordination. We conduct an evaluation with real human teammates, and the results show that **GAMMA** consistently improves performance, whether the generative model is trained on simulated populations or human datasets. Further, we propose a method for posterior sampling from the generative model that is biased towards the human data, enabling us to efficiently improve performance with only a small amount of expensive human interaction data. [1]

## 1 Introduction

Producing agents that can cooperate well with unseen partners such as humans [28] is an important problem for a variety of multi-agent systems across domains like robotics or software agents. While being cooperative is a notable characteristic of human intelligence [7, 32], training an artificial agent that cooperates well with humans poses a significant challenge. Human behaviors are uncertain and diverse, encompassing a wide range of preferences, abilities, and intentions. While humans can rapidly adapt to different partners, AI agents are particularly poor at generalizing to working with a novel human partner. Solving this issue of distribution shift between the human player's strategy and those seen during the training of an AI agent is crucial to achieving human-AI cooperation.

It is tempting to tackle this problem of learning to cooperate with humans using only data of human-human interactions [1]. While this may certainly provide some leverage, in many situations the amount of human-human data available is far less than the amount of data required by data-hungry sequential decision making algorithms. This suggests that we need a way to generate synthetic data at

---

[1]See our website for human-AI study videos and an interactive demo. The training code is also available.

38th Conference on Neural Information Processing Systems (NeurIPS 2024).

scale, while still providing relevance to the problem of coordinating with new, real human teammates on deployment. On the opposite end is the paradigm of self-play, that generates entirely synthetic data by iteratively training agents against copies of themselves. Self-play, while initially envisioned as a paradigm for learning how to *compete* with novel human players [27, 33], has been applied to cooperative multi-agent reinforcement learning to find joint strategies for a team of agents when all agents are trained together [10, 23]. Self-play is an appealing approach for learning cooperative strategies, since it can be done in simulation, and does not require the expensive and time-consuming process of collecting and training on human cooperation data [1]. However, agents trained with self-play may fail to coordinate effectively with novel humans [1, 29]. During self-play, the agent learns to form a convention with a copy of itself and relies on that convention to achieve seamless cooperation [6]. When a human adopts a different strategy at test time, the agent fails to adapt, and merely continues to follow the convention formed in training. Such scenarios are fairly common, as self-play often generates "non-human" conventions [1, 11]. This points to the dual problems of 1) human-only data being expensive, and 2) synthetic-only data lacking coverage over human behaviors.

To tackle these challenges, a popular approach to solving the distribution shift problem in human-AI cooperation has become training a Cooperator agent against a *population* of simulated agents rather than just itself [2, 15, 18, 19, 24, 29, 37, 40]. The goal of this work is often to create a diverse enough population of agents to cover the strategy space used by humans [2, 24, 37]. However, the bottom line remains that truly covering the space of strategies with discrete samples of strategies can quickly become untenable. As we move towards more complex real-world tasks, with a myriad of possible strategies, representing every strategy by an individual agent in the population and coordinating with them becomes computationally difficult. Indeed, we empirically compare the two lines of work on using simulated versus human data for zero-shot coordination, and find that contrary to past results, using human data provides better performance, especially for more complex environments.

The key insight we will make in this work is that generative models offer a solution to the dual problems mentioned above. By training a generative model that can simulate cooperation partners, we can easily incorporate both human and simulated data. The resulting model can generate a diverse range of partner strategies, by interpolating between the policies it has been trained on, as well as composing strategies to create novel partners. Therefore, we propose to *train a generative model of partner behavior using a variational autoencoder (VAE) [12]* on a collection of coordination trajectories, either human or synthetically generated. The inference model of the VAE can be used to infer the latent variable $z$ of a partner from its interaction data, encoding information about that partner's unique style or skill levels. The decoder can be used to generate the corresponding actions taken by the partner. Because the generative model can be trained on any combination of synthetic or human data, it can overcome the challenges of data scale and coverage to train adaptive Cooperators that much better cover the space of human behavior (see Figure 1).

Given the ability to generate this diverse distribution of partner policies, a single adaptive Cooperator can be trained to adapt to a range of partners, by sampling a novel partner strategy from the generative model at each episode during training. We call our full method **GAMMA**: **G**enerative **A**gent **M**odeling for **M**ulti-agent **A**daptation. While the interpolation properties of the generative model enable **GAMMA** to train more robust Cooperator agents, the availability of a controllable latent encoder also allows our desired adaptation behavior to be targeted to real human behavior. We propose a novel, ecnonomical way of incorporating a small amount of human-provided data into the sampling procedure of the generative model. This Human-Adaptive sampling method enables training Cooperators that are more targeted to partner with real human coordinators.

We test our method using Overcooked [1], a collaborative multiplayer cooking game requiring close coordination between two partners to successfully prepare recipes. In an evaluation in which trained agents play against novel humans in real-time, we find that **GAMMA** improves the performance of state-of-the-art coordination methods which rely on simulated populations of agents [24, 29], or on human data [1]. We summarize our contributions below:

- We present **GAMMA**, a novel approach for learning a generative model of partner strategies which can be trained on data from humans or a simulated population of agents. We sample from the generative model to simulate novel partners that are used to train a robust Cooperator agent. See Section 4.1 and Section 4.2.

- We propose a data-efficient and human-adaptive sampling technique to steer the model to generate more human-like partners, even with limited amounts of human data. See Section 4.3.

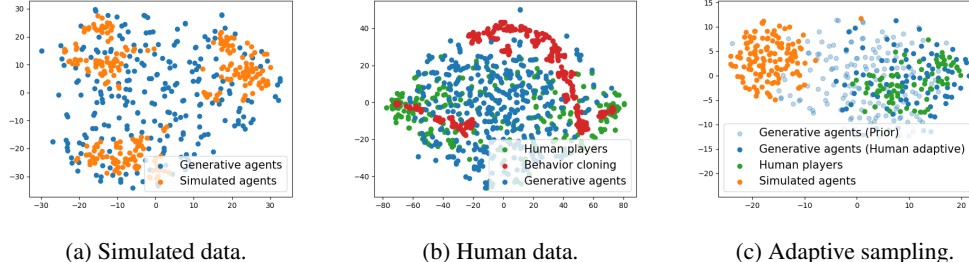

| (a) Simulated data. | (b) Human data. | (c) Adaptive sampling. |

Figure 1: We show the latent space covered by different methods.[2] For either simulated data or human data, the generative agents produced by **GAMMA** can cover a larger strategy space. Generative models can provide novel agents by interpolating the agents in the simulated population (a). On human data (b), the human proxy model only covers a subset of all human player behavior patterns, while the generative model can capture the diversity in the data. We can also control the latent space sampling (c) to model a target population of agents (e.g., human coordinators).

- We conduct an empirical evaluation via a user study with real human players. We bring together and directly compare two lines of prior work on zero-shot coordination with humans: training on simulated population data and training on human data. We find that contrary to past published results, training on human data is highly effective in our human evaluation, especially for the new, more complex layout we propose. However, regardless of the data source, **GAMMA** consistently provides significant performance improvements over multiple lines of past work when evaluated with real humans. See Section 5.

## 2 Related Work

**Training against simulated populations.** Building cooperative agents that generalize well to humans and novel partners is a long-standing problem of AI, and is known as ad-hoc team-play [28], or zero-shot coordination [11]. Recently, FCP [29] adopted the idea of using a population of policies to train a strong zero-shot Cooperator agent, as a diverse population effectively prevents the Cooperator from exploiting one specific convention. Following this framework, many techniques [2, 37] have been proposed to improve the diversity of the population, such as Maximum Entropy Population (MEP)[40]. Reward shaping [30, 37] and quality diversity [22, 31, 35] utilize domain knowledge to create agents with diverse behaviors. Statistical diversity based on trajectory distributions [19] or population entropy [40] are straightforward optimization objectives. However, as indicated in [2], agents with different behavior distributions do not necessarily use different high-level strategies. To address this, the LIPO algorithm [2] instead tries to minimize the cross-play reward between different agents in the population to encourage them to form incompatible conventions. However, because LIPO agents can potentially sabotage the game, follow-up works [3, 24] propose strategies for preventing this, including Cross-play Optimized, Mixed-play Enforced Diversity (CoMeDi) [24]. While this extensive literature presents different population creation methods, in contrast we propose a novel approach to *modeling* this population with a generative model, and thus benefit the training process through the ability to flexibly sample unlimited new partners. Our experiments directly compare to FCP [29], CoMeDi [24], and MEP [40], and show that **GAMMA** can enhance performance above each of these population-generation techniques, even using the same underlying population of simulated agents, while also allowing for the incorporation of human data.

**Training with human data.** Another line of work has focused on collecting and training on human cooperation data [1]. Learning from human data is inherently challenging due to limited scale, arbitrary human behavior, subjective preferences, and relative speed of human actions compared to agents. Another challenge of coordination comes from the heterogeneity, uncertainty and suboptimality [8, 21] of humans. The agent might fail to coordinate well when it does not properly infer the preferences or intentions of humans. Our work takes a novel approach of combining these two lines

---

[2] Points are the latent vectors. Taking the population of simulated agents (e.g., the FCP population with 8 FCP agents) as an example, a point is generated by: 1) sampling an agent in the FCP population; 2) using the agent to generate an episode with self-play; 3) using the VAE encoder to encode the episode into a latent vector and mapping the latent vector to the 2D space using t-SNE.

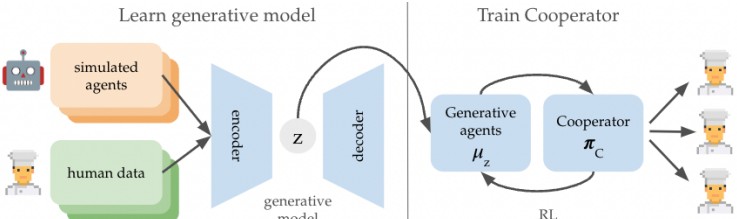

Figure 2: Overview of the method for **GAMMA**. The generative model learns a latent distribution over partner strategies from either simulated or human data. Sampling partners from the generative model enables training a robust Cooperator that can coordinate with a variety of different humans.

of work by training a generative model of partner strategies on simulated data, then using a limited amount of human data to fine-tune the model and sample from it more effectively.

**Inferring the partner type.** Our work is also related to the many works on multi-agent learning which attempt to learn a latent partner embedding to aid coordination. Some works use Bayesian inference [26, 36] or weight update [39] to decide the partner type from historical experience. Their methods require a manually-designed or predefined set of all possible partner policies. The idea of embedding-conditioned agent modeling is used in imitation learning [14] to learn more interpretable policies from human demonstrations. Grover et al. [5] train a latent variable model to learn agent representations from interaction data of different agents, and then infer a cooperation partner's latent vector and use it to condition a multi-agent RL policy. Papoudakis et al. [20] learn to predict the partner's representation from only the agent's own observation, avoiding accessing the global states during execution time. The idea of partner modeling has also been extended to offline RL [9] to train a zero-shot RL agent from a dataset. It is also known that generative models can learn diverse cooperative strategies from human-human cooperation demonstrations [34]. While our work also infers hidden context about partner type, our aims are different; we use a generative model to simulate novel partners during training time as a way to train a Cooperator to be robust for zero-shot coordination with real humans.

## 3 Problem Formulation

A multi-agent system is typically modeled as a Markov game [17]. In this work, we focus on two-player Markov games, but the formulation and our methods can be easily extended to more agents. The state space is $\mathcal{S}$. At each step, two agents execute their actions $a_t \in \mathcal{A}, b_t \in \mathcal{B}$ from their policies $\pi : \mathcal{S} \to \Delta(\mathcal{A}), \mu : \mathcal{S} \to \Delta(\mathcal{B})$. They then receive a shared reward $r_t : \mathcal{S} \times \mathcal{A} \times \mathcal{B} \to \mathbb{R}$. The next state is determined by a transition function $\mathcal{T} : \mathcal{S} \times \mathcal{A} \times \mathcal{B} \to \Delta(\mathcal{S})$. The value function $V(\pi, \mu)$ of two policies is defined as the expectation of discounted cumulative reward $\sum_{t=0}^{T} \gamma^t r_t$. Under this formulation of Markov games, the learning objective for the fully cooperative joint policy $\max_{\pi, \mu} V(\pi, \mu)$ can be perfectly defined. However, it is unclear how to describe the learning objective of human-AI cooperation under the framework of Markov games.

Inspired by the latent Markov Decision Process [13], we model the problem of cooperating with humans as an MDP with latent variable $z$. We assume the human policy is conditioned on a latent variable $z \in \mathcal{Z}$. This latent variable $z$ can be regarded as a representation of the partner's unique style, skill level, reaction speed, etc. Then the human population can be defined as a distribution $\mathcal{D}(\mathcal{Z})$ over the latent space. Given the human policy $\mu : \mathcal{S} \times \mathcal{Z} \to \Delta(B)$ conditioning on the latent variable $z$, the conditional transition function $\mathcal{T}_z(s' \mid s, a) = \sum_{b \in \mathcal{B}} \mu_z(b \mid s) T(s' \mid s, a, b)$ and the reward function $r_z(s, a) = \sum_{b \in \mathcal{B}} \mu_z(b \mid s) r(s, a, b)$ can both be defined. Now with the value function for MDP $M_z = \{\mathcal{T}_z, r_z\}$ is $V_z(\pi) = V(\pi, \mu_z)$, the learning objective can be defined as $\pi^* \in \arg\max_\pi \mathbb{E}_{z \sim \mathcal{D}(\mathcal{Z})} [V_z(\pi)]$.

## 4 GAMMA: Generative Agent Modeling for Multi-agent Adaptation

We will leverage generative models as our key tool for strategy diversification, to overcome the dual challenges of 1) lack of real human data and 2) the difficulty of synthetically covering the large strategy space of human partners. In being able to *generate* diverse strategy profiles beyond the data,

these generative models can be used to train a Cooperator that can be deployed to coordinate with novel users, each with their own diverse styles, preferences and capabilities. Here, we first describe our procedure for learning a generative model from training data of discrete agents (or human-human data), and then describe how this can be used for targeted coordination with real human partners. We call our approach **G**enerative **A**gent **M**odeling for **M**ulti-agent **A**daptation (**GAMMA**).

## 4.1 Learning Generative Models of Partner Behavior

The first step of **GAMMA** is learning the generative model from pre-existing coordination data. We assume access to a dataset of trajectories $\mathcal{D}_{\text{coordination}} = \{\tau_i\}_{i=1}^{N}$, where each trajectory $\tau = \{(s_t, a_t, b_t)\}_{t=0}^{T}$ is a sequence of multi-agent coordination behaviors. This dataset can be derived from human playing records. Alternatively, if there is a population of simulated agents available, this dataset can be collected by pairing them together to generate joint trajectories. For instance, in our experimental evaluation, we use agents generated through techniques like fictitious co-play (FCP) [29] to generate this dataset. The purpose of generative modeling here is to be able to model the multimodal marginal distribution over various strategy profiles in $\mathcal{D}_{\text{coordination}}$, a challenging distribution to model with standard maximum likelihood methods. Notably, the generative model is able to sample a landscape of agents that can be used to train a Cooperator, going beyond the quantity and diversity of the training data.

We develop a variant of the Variational Autoencoder (VAE) [12] to model the diverse strategies underlying the training data $\mathcal{D}$. As is typical in variational inference frameworks, we propose to learn an encoder-decoder generative model with an approximate posterior (encoder) $q(z \mid \tau; \phi)$ that identifies the agent style from the trajectory. The decoder in this model, $p(a_t \mid z, \tau_{0..t-1}; \theta)$ uses the agent's own past experience $\tau_{0..t-1} = (s_0, a_0, ..., s_{t-1}, a_{t-1}, s_t)$ and the latent variable $z$ to predict the agent's next action. This structure of the variational inference model can be thought of as multi-modal behavior cloning, where the encoder history provides the latent variable required to model the distribution of generated actions. The encoder-decoder architecture described above, and shown in Figure 2 can be trained using an evidence lower bound (ELBO) loss:

$$\mathcal{L}(\theta, \phi) = \mathbb{E}_{\tau \sim D} \left[ \mathbb{E}_{z \sim q(\cdot \mid \tau, \phi)} \left[ \sum_{t=1}^{T} \log p(a_t \mid z, \tau_{0..t-1}; \theta) \right] + \beta \text{KL}(q(z \mid \tau, \phi) \parallel \mathcal{N}(0, I)) \right] \quad (1)$$

Importantly, this generative model is able to generate behaviors that go far beyond the training data, both in quantity and diversity, as is shown in Figure 1. We show in Fig 1 that while simulated behavior may not cover the space by itself, the generative model has significantly greater coverage over the strategy space. Importantly, we hypothesize that human behavior is more likely to lie within the span of strategies generated by the generative model. In this way, the interpolation and generalization of the generative model (even if it is not perfect) provides the expansion of the data required to effectively coordinate with humans.

## 4.2 Training a Cooperator with Generative Coordination Models

Once a generative model has been obtained, it can be used to train a robust Cooperator agent. This is accomplished by treating the generative model as a partner generator, using it to simulate partner behavior that covers the space of real human behaviors, and training the Cooperator to optimize performance with these partners in simulation. In particular, the generative agent model $p(a_t \mid z, \tau_{0..t-1}; \theta)$ can now be used as a generator of partner policies $\mu_z$ to train our Cooperator agent. As shown in Figure 2, for every episode we can sample latent variables from the prior $z \sim p(z)$, and use the conditioned action distribution $p(a_t \mid z, \tau_{0..t-1}; \theta)$ as a partner $\mu_z$ for that episode. We aim to leverage these sampled partners from the generative model to train a single Cooperator $\pi_C$, treating the sampled partners $\mu_z$ as part of the environment. At each iteration, a batch of agents $\{\mu_{z_i}\}_{i=1}^{N}$ are generated using latent variable samples $z_i \sim D_z$. Then a batch of MDPs $\{M_{z_i}\}_{i=1}^{N}$ can be derived from these training partner policies to learn our Cooperator policy $\pi$. Then we use PPO [25] to optimize $\pi_C$ over these training MDPs:

$$J(\pi_C) = \mathbb{E}_{z \sim \mathcal{D}_z} [V_z(\pi)] = \mathbb{E}_{z \sim p(z)} [V(\pi, \mu_z)]. \quad (2)$$

Importantly, the Cooperator $\pi_C$ is not trained with imitation learning, but is rather trained with reinforcement learning to learn task-specific coordination behavior. The generative model from

Section 4.1 is simply used to provide the quantity and diversity of agents necessary for meaningful generalization to real human behavior.

### 4.3 Targeted GAMMA using Human-Adaptive Sampling and Fine-tuning

While the Cooperator described in Section 4.2 is trained by sampling partner agents from the generative model as $z \sim p(z), a_t \sim p(a_t \mid z, \tau_{0..t-1}; \theta)$, this does not make use of human specific data if available, instead simply treating any human and synthetic data as equivalent. However, when coordinating with real human partners, it is useful to target model adaptation to be human-specific rather than to span the entire space of potentially irrelevant synthetic strategies (as shown in Figure 1). The key insight we will leverage to do so is that the latent space afforded by our generative model provides the ability to do *controllable sampling* from any latent distribution. This distribution can be chosen to incorporate additional information about the desired population. In particular, when coordinating with real human partners, the latent prior distribution for sampling $p(z)$ can be replaced with a human-centric one $p_h(z)$, which is more focused on the part of the latent space relevant to human behavior. This suggests that given a small amount of human data $\mathcal{D}_h$, a human-centered latent distribution $p_h(z)$ can be quickly inferred by encoding the human data and estimating the latent Gaussian distribution that best explains encoded human data. The latent Gaussian mean is given by:

$$\bar{z} = \mathbb{E}_{\tau_h \in D_h} \left[ \mathbb{E}_{z \in q(\cdot | \tau_h)}[z] \right]. \tag{3}$$

Given this human-centered latent distribution $p_h(z) = \mathcal{N}(\bar{z}, I)$, a targeted Cooperator that is meant for a particular target population can be trained by maximizing $J(\pi_C) = \mathbb{E}_{z \sim p_h(z)} \left[ V(\pi, \mu_z) \right]$.

Human-adaptive sampling makes the model focus less on adaptation to irrelevant synthetic partners and more on "human-centric" partners sampled from the generative model. As our experiments will show, this approach outperforms training a partner simulator using only human data, since with limited human data it is easy for the model to go out of distribution during generation and it can thus be brittle. In contrast, **GAMMA** is able to train a robust partner simulator using large amounts of synthetic data. Adding human-adaptive sampling to **GAMMA** can provide considerable generalization benefits even with modest amounts of data. Note that to better capture human data, we do not just target the latent space, but also perform some fine-tuning on the encoder and decoder of the VAE using human data.

## 5 Experiments

We evaluate **GAMMA** using the Overcooked environment [1] as a popular benchmark for prior work on human-AI cooperation [1, 24, 29, 36, 37]. In Overcooked, two players need to cooperate to divide the work of cooking and avoid getting in each other's way. This involves anticipating the intended goal and actions of the other player, and inferring which task would most usefully assist them. The layouts proposed in previous work are shown as the first five environments in Figure 3. The *Counter Circuit* layout poses a hard coordination challenge with multiple strategies such as passing items across the countertop to the partner or moving clockwise/counterclockwise around the ring. Additionally, we include a custom layout based on *Counter Circuit*, termed *Multi-strategy Counter*. This new layout involves recipes with multiple ingredients, significantly increasing the complexity of the strategy space. Failing to infer the intentions of the partners and adding the wrong ingredients to the pot could ruin the entire dish, which increases the importance of coordination.

Our experiments investigate the following questions:

**H1: Using simulated agents.** Will training a Cooperator against a generative model of partner strategies trained on a *population of simulated agents* outperform training the Cooperator against the simulated agents directly?

**H2: Using real human data.** Can the generative model be used to effectively leverage (small amounts of) human data, and also combine it with simulated agents?

**H3: State-of-the-art.** Can we obtain better performance than competitive baselines using both simulated and human data?

**Learning from a Simulated Agent Population.** We first investigate H1, and test whether generative agents can improve the performance of the Cooperator agent when training with simulated agent partners. We include three state-of-the-art approaches for creating a population of simulated agents.

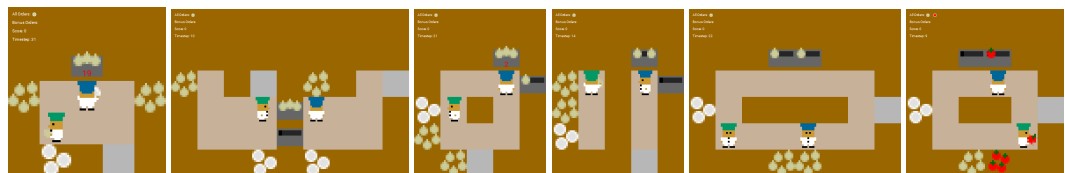

Figure 3: The first five layouts *Cramped Room, Asymmetric Advantages, Coordination Ring, Forced Coordination, Counter Circuit* are originally proposed in Carroll et al. [1]. We create an additional *Multi-strategy Counter* layout. In this new layout, humans can additionally choose between making onion vs. tomato soup, which makes coordination significantly more challenging.

**Fictitious Co-Play (FCP)** [29] uses multiple self-play agents initialized across different random seeds with checkpoints sampled throughout training, which simulates a diverse population with varied skill levels. **Maximum Entropy Population-based Training (MEP)** [40] enhances FCP by further diversifying the population through a population entropy learning objective. **Cross-play Optimized, Mixed-play Enforced Diversity (CoMeDi)** [24] generates a diverse set of coordination policies by minimizing the cross-play reward and prevents self-sabotage using mixed-play regularization. For each method, we use the simulated agent population to create a dataset to learn the generative model.

**Learning from Human Data.** To investigate H2, we test whether the generative model can be used to model human behaviors from data produced by real human players. In this work, we assume the human dataset contains 20 to 50 trajectories, reflecting the popular open-source dataset provided by Carroll et al. [1]. We use **PPO-BC** [1] as the baseline where a BC model over human data is used as the partner during training. We test whether replacing the BC agent in this framework with our generative model trained on the human data (**PPO-BC-GAMMA**) provides better performance. Since we assume the amount of human data is limited, we also fine-tune a generative model pretrained from the simulated agent population with human data with both decoder-only (DFT) and full fine-tuning (DFT), as described in Section 4.3. We apply our **Human-Adaptive (HA)** sampling method to the fine-tuned generative model to sample from the human-centered latent distribution $p_h(z)$.

**Simulated Agent Evaluation.** Following prior work [1], we create an automatic evaluation mechanism by using held-out human data to train a behavior cloning policy as a human proxy agent, and report the performance of the Cooperator when it is paired with this test human proxy agent. However, we note that methods which are explicitly trained against a human-proxy agent (PPO-BC) can easily exploit this metric in a way that is not indicative of actual performance with real humans, so we do not use this automatic evaluation to assess those methods. Instead, we conduct evaluations with real human players as the gold standard evaluation technique.

**Human Evaluation.** We run a user study with real human players in order to determine which method can most effectively coordinate with humans, and which method is rated as subjectively better by humans (H3). We conducted a study with 80 users recruited via online crowdsourcing from Prolific. Our study follows guidelines set by a UW IRB protocol. During the study, each user is instructed to play multiple rounds of Overcooked with a partner via a web interface, where in each round the partner is an agent following one of the 9 policies, in randomized order. We trained 5 random seeds per agent, and used a different randomly-selected seed for each of the $9PP$ game rounds. For the human study, we focus on the most complex layouts, *Counter Circuit* and *Multi-strategy Counter*. Each game lasts for 60 seconds. After each round, the user answers Likert scale survey questions [16] to rate their experience playing with the agent. At the end of the 8 rounds, humans also answer qualitative questions about the performance of the agents. Users are required to pass an attention check to ensure quality of their data. Using these answers, we conduct a qualitative analysis to understand which factors most heavily influence overall performance and users' preferences when playing with real humans.

# 6 Results

## 6.1 Evaluation in Simulation

In this section, we present the evaluation results against a human proxy (behavior cloning) agent.

**H1: Using Simulated Data.** We train the generative models **FCP+GAMMA**, **CoMeDi+GAMMA** and **MEP+GAMMA** on the simulated agent population of FCP, CoMeDi and MEP, respectively. Figure 4 shows the learning curves of each method averaged on all layouts, and the final performance

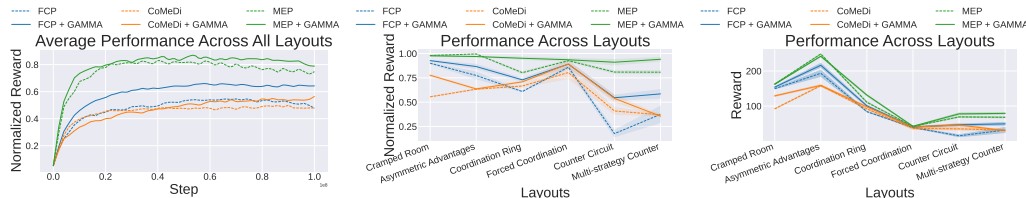

(a) Learning curves of different methods.  (b) Normalized reward across different layouts.  (c) Original reward across different layouts.

Figure 4: Evaluation of different methods using a human proxy model. Rewards are normalized by the highest reward achieved on each layout. The learning curves in (a) show the average normalized reward across all environments, indicating that GAMMA helps the Cooperator converge to a higher reward. This improvement is also consistent across individual layouts, as illustrated in (b) and (c). We observe the largest performance gap on the 'Counter Circuit' and 'Multi-Strategy Counter' layouts, which are the most complex in terms of the number of valid cooperation strategies.

of different methods for each of the different layouts. The Cooperator agent is evaluated by the zero-shot cooperation performance with a human proxy BC model. **GAMMA** consistently improves performance over the baselines, across layouts. This demonstrates that **GAMMA** provides a more efficient way to utilize the simulated agents by providing a landscape of partners to train the Cooperator. We also find the improvement gap increases as the layouts become more complex.

## 6.2 Evaluation with Real, Novel Human Partners

As described in Section 5, we conduct an evaluation with real human players, recruiting new participants that were not in the training data. We evaluate all agents against the novel human players, and plot the cooperative scores achieved by each agent-human team in Table 3.

### 6.2.1 H1: Training with Simulated Data.

We find that **GAMMA** offers significant improvements over prior techniques for training against simulated populations (**FCP, CoMeDi, MEP**). Although the trend of these results is consistent with the previous results for H1, we find that here **GAMMA** provides significantly enhanced performance improvements when tested with real humans, reaching the new state-of-the-art performance for both *Counter Circuit* and *Multi-Strategy Counter*.

On our newly proposed, more complex layout, *Multi-Strategy Counter*, we find that the CoMeDi baseline performs so poorly that it cannot discover strategies which make use of the new tomato ingredient. The reason is because playing the onion-soup strategy and the tomato-soup strategy together can recreate an unrecoverable game state, which is not favored by the CoMeDi algorithm to include both types of agents in the population. Therefore, when we use **GAMMA** to train a generative partner model using data generated from the **CoMeDi** population, it also fails to learn any strategies involving tomatoes, and both models generalize poorly to playing with humans, although **CoMeDi+GAMMA** still offers a performance benefit over **CoMeDi**. This points to the fact that **GAMMA**, which is based on training a generative model on cooperation data, can fail to perform well if the cooperation data is not sufficiently diverse. This problem can be aptly described by the well-known adage "*garbage in, garbage out*".

However, when **GAMMA** is trained using the high-quality population found by **MEP**, we see that it performs extremely well, reaching a score of 88 on *Multi-Strategy COunter*, while MEP, the most competitive simulated population baseline, only achieved 64. This represents a 38% improvement.

### 6.2.2 H2: Training with Human Data.

Comparing methods that make use of human data reveals some interesting findings. First, our results directly compare two lines of prior research: training on simulated populations vs. training against a BC model trained with a limited amount of human data (PPO-BC). While previous works show that using only simulated data can exceed PPO-BC and reach state-of-the-art performance [29], we find that on the more complex layout, PPO-BC is actually the best performing baseline. Modeling

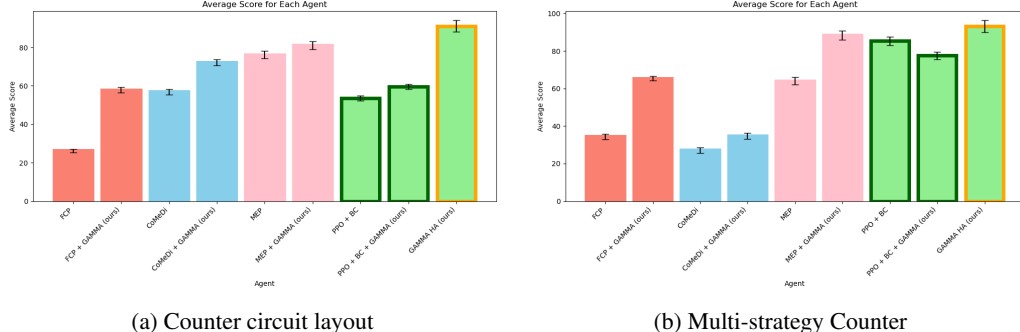

(a) Counter circuit layout          (b) Multi-strategy Counter

Figure 5: Performance of different agents when played with real humans. Error bars [4] use the Standard Error of the Mean (SE) for statistical significance ($p < 0.05$). Methods trained on human data are shown in green. Whether training with simulated or human data, **GAMMA** shows consistent, statistically significant advantages over the baselines. **GAMMA-HA** is able to efficiently use the real human dataset to learn a better sampling of its latent space, achieving the best performance when cooperating with real humans.

the human data with the generative model (**PPO+BC+GAMMA**) only provides performance improvements half the time. However, combining simulated and human data with **Human Adaptive GAMMA** provides significantly higher performance in both layouts, surpassing state-of-the-art zero-shot coordination techniques.

### 6.3 H3: Can we obtain better performance than competitive baselines?

As revealed in Figure 5, **GAMMA HA** achieves the highest performance in both layouts, surpassing the most competitive baselines by 60% and 43% in *Counter Circuit* and *Multi-Strategy Counter*, respectively (See Table 3). We conducted a statistical analysis of these results using Holm-Bonferroni correction, which can be found in Table 4.

#### 6.3.1 Subjective Human Ratings

As demonstrated in prior work [1], humans can adapt to deficiencies in the policies of AI agents and narrow the apparent performance gap between different agents by simply completing the task themselves. This means cooperation performance alone cannot measure whether a particular agent is frustrating or cumbersome for the human to cooperate with. Therefore, we also collect subjective ratings of the agents from the human participants. The results for the question about overall cooperation are plotted in Figure 6; additional results are available in the Appendix, which pertain to agent's ability to adapt (Fig. 9), and whether it was human-like (Fig. 10) or frustrating (Fig. 11).

For the overall ratings in Figure 6,the subjective ratings mirror the cooperative performance scores: when **GAMMA** is trained with the same data available to a baseline technique it improves performance in terms of the human ratings, and **GAMMA HA** gives consistently good performance on both layouts, in the sense that it receives a higher proportion of positive human ratings. We note that while PPO+BC+**GAMMA** did not show a performance benefit over PPO+BC on *Multi-strategy Counter*, human ratings of PPO+BC+**GAMMA** were more positive. One exception to the previous trends is that CoMeDi obtains high subjective ratings in *Counter Circuit*, although it performs poorly in *Multi-strategy Counter*. From the additional figures in the appendix, we can observe that **GAMMA** methods are rated as more adaptive, human-like, and less frustrating than other techniques.

#### 6.3.2 Qualitative Findings

Analyzing the qualitative results reported by participants in the human study, we find that **adaptation to the human partner** was a core theme distinguishing agents that humans liked. Participants reported that the **FCP + GAMMA** agent demonstrated an ability to learn from the user's actions, such as mimicking the user's strategy of placing onions on the table to save time. This adaptive behavior was positively received by a study participant: "I noticed that once I started to put back onions on the table that it did the same as I wanted to save time rather than going back for onions 3 times for soup. I thought it was interesting that it learned about my behavior." Because **GAMMA** models have been trained over a more diverse range of partners, they consequently exhibit the ability to better adapt to real humans during gameplay.

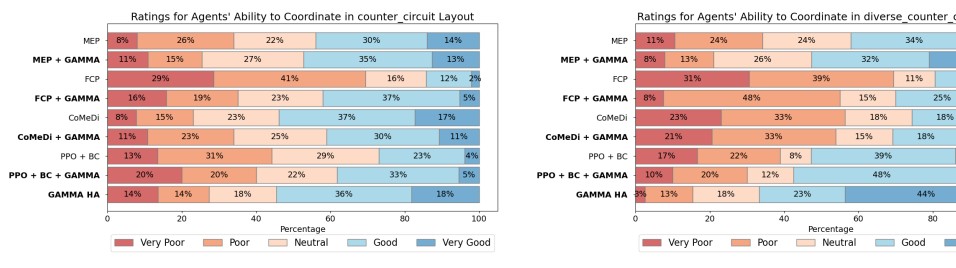

| | | | |
|---|---|---|---|
| (a) Counter circuit layout | | (b) Multi-strategy Counter | |

Figure 6: Human ratings for different agents. Individuals were asked to respond to the following question: "Overall, I felt that the agent's ability to coordinate with me was: {Very poor, Poor, Neutral, Good, Very good}". **FCP + GAMMA**, **PPO + BC + GAMMA**, and **GAMMA-HA** consistently receive higher ratings for ability to coordinate compared to their respective baselines.

Another common theme that emerged from the qualitative analysis was **consistency and predictability.** Erratic behavior was a common observation regarding the baseline methods. Users observed the AI performing random actions, such as moving onions around without an evident purpose or failing to complete necessary steps in the cooking process. In contrast, users report that agents using **GAMMA** behave in a logical, consistent manner. For example, for **FCP + GAMMA**, participants provided the following feedback that the agent was "more deliberate in its actions" and "its actions were logical". In contrast users reported that the baseline FCP agent, "didn't behave logically" and "the agent this time was inconsistent and did not help me with any of my orders at all." We hypothesize the reason for this "inconsistency" that occurred in baseline methods may be due to the human's behavior going out-of-distribution (OOD) of their training data, causing the resulting policy to make errors and behave in an unpredictable way. This points to the importance of obtaining better coverage of the human data distribution, as provided by **GAMMA** (see Figure 1).

## 7 Conclusion

In this work, we propose **GAMMA**, a novel approach to training a coordinator agent by using generative models to produce training partner agents. We conduct a comprehensive analysis using data from a study with real human cooperation partners, and show **GAMMA** outperforms baselines over both subjective human ratings and quantitative measurements of cooperation performance. We also provide a new perspective to compare different populations under the latent space of a generative model, showing how the simulated populations may not provide sufficient coverage of the range of human players.

**Limitations.** As shown by the performance of GAMMA+CoMeDi on *Multi-Strategy Counter*, obtaining good performance with our approach depends on having a reasonably diverse amount of cooperation data to train the model. If the quality of the simulated population data is too low, the approach can fail to provide significant benefits.

In this work, our human studies recruit participants from Prolific, which may not be representative of broader populations. Additionally, our human dataset is limited, which could reduce the diversity of strategies and force participants to adapt to strategies that the Cooperators are already familiar with.

We focus on the two-player setting in this study following prior work [29, 37, 40] because it is a first step toward enabling an AI assistant that could help a human with a particular task. Scaling up to more agents would exponentially increase the dataset size with our current techniques. Therefore, better sampling techniques are needed to address this issue.

**Future work.** Several potential directions are interesting for future work: 1) In this work, the amount of human data is limited, which restricts the performance of the generative model that learns human data from scratch. 2) An orthogonal direction is to condition the policy of the Cooperator on the embedding of the partner policy. We provide some preliminary results in Figure 15.

**Social impact.** Our work focuses on how to train AI agents that can effectively cooperate with diverse humans to assist them with tasks. We believe this is a critical component of eventually enabling assistive robots that could operate in human environments to assist the elderly or disabled to live more comfortably, or reduce the burden of domestic labour for all people.

## Acknowledgment

SSD acknowledges the support of NSF IIS 2110170, NSF DMS 2134106, NSF CCF 2212261, NSF IIS 2143493, NSF CCF 2019844, and NSF IIS 2229881.

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

## A  Reproducibility

Our demo website is https://sites.google.com/view/human-ai-gamma-2024/ and contains the code and more experiment results. We also provide information about the implementation details B and hyperparameters used in our experiments D to help reproduce our results.

## B  Implementation details

**Generative models.** The dataset used to train the VAE model contains the joint trajectories of two players. For the simulated agent population $\{\pi_1, ..., \pi_N\}$, we create this dataset by evenly sampling $\pi_i \times \pi_j$ to generate the trajectories. A simulated dataset contains 100K joint trajectories. To train a VAE on it, the dataset is split into a training dataset with 70% data and a validation dataset with the rest of 30% data. To compute the ELBO loss 1, the trajectories are truncated to length 100 for better optimization for the recurrent module. A linear scheduling of the KL penalty coefficient $\beta$ is adopted to control a target value for the KL divergence of the posterior distribution. The target KL value is set to 7 for the layout *Forced Coordination* over the CoMeDi population. All other VAE models are chosen by a target KL value of 32.

**Train Cooperator agents.** On Overcooked, the Cooperator is trained by PPO [25]. We based our implementations on HSP [37]. Reward shaping for dish and soup pick-up is used for the first 100M steps to encourage exploration. All results are reported with averaged episode reward and the standard error of at least 5 seeds.

**Simulated agent populations.** We use MAPPO [38] to create the FCP [29] agent populations. Eight self-play agents are trained and three checkpoints for each agent are added to the population, making the population size $8 \times 3$. For the CoMeDi agent population [24], we download the population proposed by the authors [3] for the original five layouts [1]. The population size is 8 for CoMeDi. For our custom layout *Multi-strategy Counter*, we use CoMeDi's official implementation and keep the population size the same.

## C  Human Dataset

For the human dataset in the original Overcooked paper [1], their open-sourced dataset contains "16 joint human-human trajectories for Cramped Room environment, 17 for Asymmetric Advantages, 16 for Coordination Ring, 12 for Forced Coordination, and 15 for Counter Circuit.." with length of $T \approx 1200$. In Multi-strategy Counter, we collect 38 trajectories with length $T \approx 400$ which is closer to the actual episode length during training.

## D  Hyperparameters

We use MAPPO to train our Cooperator agent. The architectures and hyperparamers are fixed throughout all layouts. All policy networks follow the same structure where an RNN (we use GRU) is followed by a CNN.

The generative model follows a similar architecture to the policy model. An encoder head and a decoder head are used to produce variational posterior and action reconstruction predictions from the representations.

## E  Computational Resources

We conducted our main experiments on clusters of AMD EPYC 64-Core Processor and NVIDIA A40/L40. It takes about one day to train one Cooperator agent. The main experiments takes about 3600 GPU hours. We do some preliminary experiments to search for the best hyperparameters and training frameworks.

---

[3]https://github.com/Stanford-ILIAD/Diverse-Conventions/tree/master

| hyperparameter | value |
| --- | --- |
| CNN kernels | [3, 3], [3, 3], [3, 3] |
| CNN channels | [32, 64, 32] |
| hidden layer size | [64] |
| recurrent layer size | 64 |
| activation function | ReLU |
| weight decay | 0 |
| environment steps | 100M (simulated data) or 150M (human data) |
| parallel environments | 200 |
| episode length | 400 |
| PPO batch size | $2 \times 200 \times 400$ |
| PPO epoch | 15 |
| PPO learning rate | 0.0005 |
| Generalized Advantage Estimator (GAE) $\lambda$ | 0.95 |
| discounting factor $\gamma$ | 0.99 |

Table 1: Hyperparameters for policy models

| hyperparameter | value |
| --- | --- |
| CNN kernels | [3, 3], [3, 3], [3, 3] |
| CNN channels | [32, 64, 32] |
| hidden layer size | [256] |
| recurrent layer size | 256 |
| activation function | ReLU |
| weight decay | 0.0001 |
| parallel environments | 200 |
| episode length | 400 |
| epoch | 500 |
| chunk length | 100 |
| learning rate | 0.0005 |
| KL penalty coefficient $\beta$ | $0 \rightarrow 1$ |
| latent variable dimension | 16 |

Table 2: Hyperparameters for VAE models

# F   Human Evaluation

Since diversity of humans is a key component in our approach, in this section we report the demographics of our participants. The study demographics include a population of $54\%$ female and $46\%$ male participants. The user demographics skewed towards younger ages, with $39\%$ of participants between ages $18 - 26$, $44\%$ of participants between $27 - 37$, $11\%$ of participants between $38 - 48$, and $6\%$ of participants ages $49$ and above. The game experiences of the participants are shown in Figure 7.

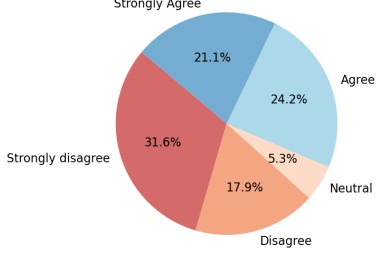

Figure 7: Percent of participants who agree with the statement "I have experience playing the game Overcooked".

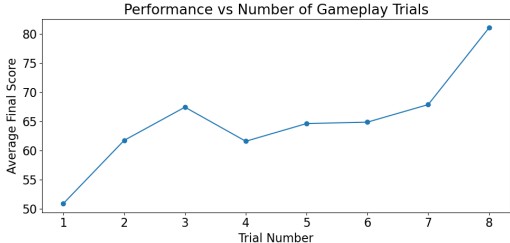

Figure 8: Human performance improves with the number of trials, indicating that the humans learn, change, and adapt their gameplay during the course of the evaluation.

## F.1  Humans are adapting during evaluation

As shown by Fig 8, human performance improves with the number of trials, indicating that the humans learn, change, and adapt their gameplay during the course of the evaluation. In our evaluation with real humans, each user can change their strategy at any point in the game. A significant proportion of our users self-identify as novice players, both for video games and for Overcooked (17.9% and 49.5%, respectively, Figure 8, thus often exhibiting improved performance over the course of an increased number of trials. Figure 8 provides evidence of this pattern, demonstrating that humans change their gameplay style over the course of the evaluation.

# G  Additional Human Study Results

## G.1  Human-AI team scores

| Agent | Training data source | Counter circuit | Multi-strategy Counter |
|---|---|---|---|
| FCP | FCP-generated population | $26.24 \pm 0.96$ | $34.35 \pm 1.47$ |
| FCP + GAMMA | | $\mathbf{57.87 \pm 1.35}$ | $\mathbf{65.36 \pm 1.24}$ |
| CoMeDi | CoMeDi-generated population | $56.87 \pm 1.43$ | $27.11 \pm 1.46$ |
| CoMeDi + GAMMA | | $\mathbf{72.17 \pm 1.61}$ | $\mathbf{34.72 \pm 1.65}$ |
| MEP | MEP-generated population | $76.19 \pm 1.89$ | $64.02 \pm 2.07$ |
| MEP + GAMMA | | $\mathbf{81.10 \pm 2.03}$ | $\mathbf{88.30 \pm 2.51}$ |
| PPO + BC | human data | $53.51 \pm 1.37$ | $\mathbf{85.26 \pm 2.28}$ |
| PPO + BC + GAMMA | | $\mathbf{59.67 \pm 1.35}$ | $77.53 \pm 2.00$ |
| GAMMA HA | MEP pop. + human data | $\mathbf{91.11 \pm 2.96}$ | $\mathbf{93.09 \pm 3.19}$ |

Table 3: Human evaluation results. Our methods (GAMMA) show significant improvements.

We conducted statistical significance tests and computed the $p$-value using the Holm-Bonferroni correction. See Table 4.

| Hypothesis | Counter circuit ($p$-value) | Multi-($p$-value) |
|---|---|---|
| FCP + GAMMA > FCP | $1.27 \times 10^{-69}$ | $1.43 \times 10^{-50}$ |
| CoMeDi + GAMMA > CoMeDi | $7.31 \times 10^{-12}$ | $3.25 \times 10^{-3}$ |
| MEP + GAMMA > MEP | $0.639$ | $2.04 \times 10^{-12}$ |
| PPO + BC + GAMMA > PPO + BC | $9.80 \times 10^{-3}$ | $7.48 \times 10^{-2}$ |
| GAMMA HA > FCP | $4.21 \times 10^{-113}$ | $1.10 \times 10^{-63}$ |
| GAMMA HA > CoMeDi | $1.55 \times 10^{-23}$ | $3.43 \times 10^{-77}$ |
| GAMMA HA > MEP | $1.43 \times 10^{-4}$ | $2.92 \times 10^{-13}$ |
| GAMMA HA > PPO + BC | $3.14 \times 10^{-32}$ | $0.426$ |

Table 4: Statistical significance ($p < 0.05$) is achieved for the majority of the results.

### G.2 Qualitative analysis

We include additional analyses of users self-reported responses for qualitative questions from our user study. Figure 9 shows the results for users' response to the agents' ability to adapt. The results indicate that two of our methods, **FCP + GAMMA** and **GAMMA-HA-DFT**, consistently receive higher ratings indicating better ability to adapt than their respective baseline agents, across both layouts. Figure 10 displays users' ratings for whether the agents demonstrated human-like behavior. The responses show that **FCP + GAMMA**, **CoMeDi + GAMMA**, and **GAMMA-HA-DFT** exhibit the most human-like gameplay in both layouts. Figure 11 includes responses for whether the agents' behavior was frustrating. Individuals consistently reported that **FCP + GAMMA** and **GAMMA-HA-DFT** demonstrated the least frustrating behavior.

Furthermore, we provide additional participant feedback for our agents from the user study as follows, starting with feedback for agents using our methods:

Feedback for **FCP + GAMMA**:

- "It was very coordinated."
- "This agent figured things out the fastest and worked with me well."
- "When it was in my way, it moved out of the way instead of blocking the path, which was an issue with other agents."
- "Much better cooperation by comparison."

Feedback for **CoMeDi + GAMMA**:

- "He was the best teammate."
- "This one did well, it moved around me enough to complete tasks and we were relatively efficient passing items around."
- "The agent figured out the next step I would have done."

Feedback for **PPO + BC + GAMMA**:

- "It was very efficient and placed things down so I could grab it on the other side."
- "This agent was very collaborative. All I did was put onions out and he did the rest of the work - super smooth."
- "It behaved smoothly and intentionally. It responded to my moves and helped rather than getting in the way."

In contrast, the feedback for the baseline agents is overall more negative, including themes such as frustration, inefficiency, and lack of cooperation.

Feedback for **FCP** (baseline):

- "Frustrating."
- "He got in the way and didn't help at all."
- "Just seemed to lack coordination and felt less intelligent."

Feedback for **CoMeDi** (baseline):

- "The agent was helpful, but still did not fully cooperate as well as I thought it should have."
- "The agent was executing some random move actions before actually cooperating."

Feedback for **PPO + BC** (baseline):

- "The agent was not helping much, it was doing its own thing, which meant we were not coordinating well."
- "Agent did everything by himself; didn't see the onions on the counter that I had placed; got in my way a bunch."
- "Very inefficient and lacking intelligence."

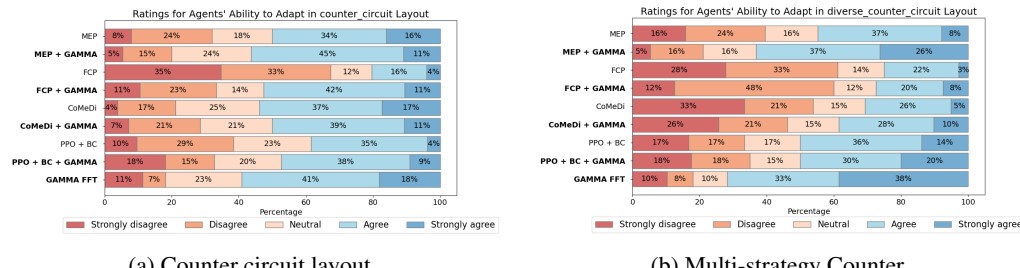

(a) Counter circuit layout                    (b) Multi-strategy Counter

Figure 9: Human ratings for different agents. Individuals were asked to respond to the following question: "The agent adapted to me when making decisions: {Strongly Disagree, Disagree, Neutral, Agree, Strongly Agree}". **FCP + GAMMA** and **GAMMA-HA-DFT** consistently receive higher ratings for ability to adapt compared to their respective baselines.

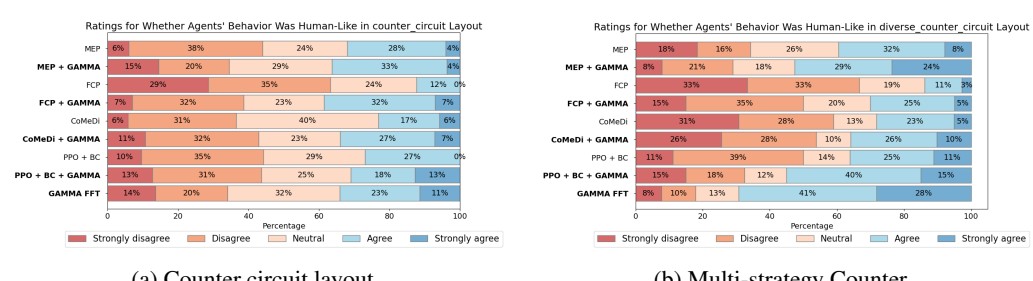

(a) Counter circuit layout                    (b) Multi-strategy Counter

Figure 10: Human ratings for different agents. Individuals were asked to respond to the following question: "The agent's actions were human-like: {Strongly Disagree, Disagree, Neutral, Agree, Strongly Agree}". **FCP + GAMMA**, **CoMeDi + GAMMA**, and **GAMMA-HA-DFT** consistently receive ratings for more human-like behavior compared to their respective baselines.

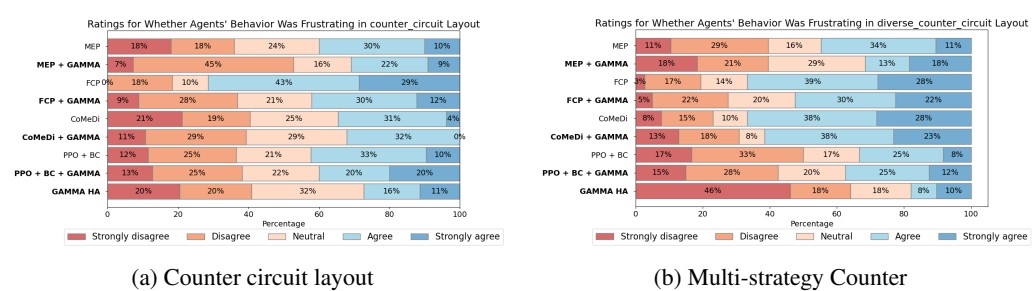

(a) Counter circuit layout                    (b) Multi-strategy Counter

Figure 11: Human ratings for different agents. Individuals were asked to respond to the following question: "The agent's behavior was frustrating: {Strongly Disagree, Disagree, Neutral, Agree, Strongly Agree}". **FCP + GAMMA** and **GAMMA-HA** consistently receive ratings for less frustrating behavior compared to their respective baselines.

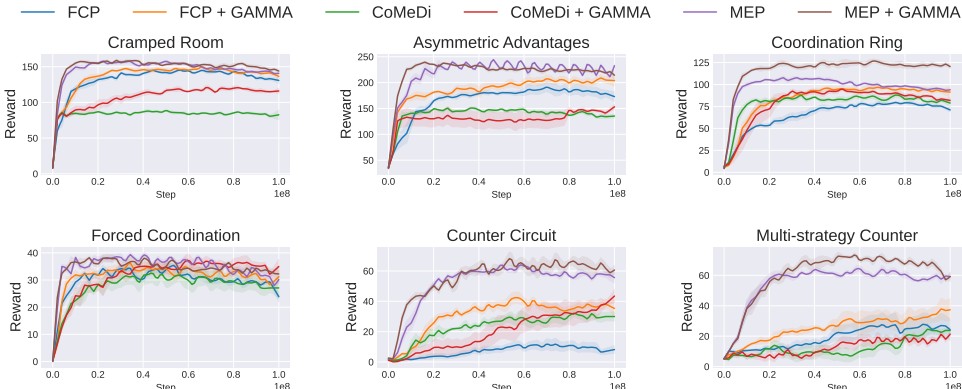

Figure 12: Learning curves for methods using simulated data across six layouts. Error bars are the Standard Error of the Mean (SE). All methods are evaluated using a held-out human proxy model as the partner player. **GAMMA** consistently shows better or equal performance on all layouts for both simulated data-generation methods (FCP, CoMeDi, MEP) when evaluated against the human-proxy model.

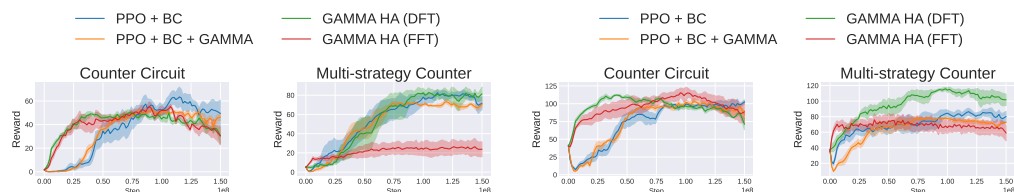

(a) Evaluation against human proxy agent.      (b) Evaluation against held-out self-play agents.

Figure 13: Learning curves for methods using human data. When evaluated with a held-out human proxy agent (a), human adaptive sampling learns faster on *Counter Circuit*, but does not reach better final performance since PPO-BC is trained to exploit a human-proxy agent. With simulated self-play partners (b), Human-Adaptive **GAMMA** with DFT shows better performance.

## H Additional simulated results

Figure 12 provides the original data for the performance on simulated data.

In addition to training the generative model on the human dataset (**PPO-BC-GAMMA**, we can also leverage Human Adaptive (HA) sampling on the generative model: (**GAMMA-HA**). The learning curves of these methods and the baseline **PPO-BC** [1] are shown in Figure 13. When evaluated with a held-out human proxy agent, our methods do not show a great advantage. This is expected, because PPO-BC is trained to exploit a human proxy agent. However, in the human study, we show that this hurts its adaptation to more diverse real human players. On the other hand, we note that **GAMMA-HA** shows much faster learning in both layouts. Figure 13b shows that **GAMMA-HA** also adapts better to held-out self-play agents compared to PPO-BC.

## I Compare decoder-only fine-tuning and full fine-tuning

### I.1 Full fine-tuning can suffer from insufficient human data

In some early experiments, we find that only fine-tuning the decoder with human data (**GAMMA-HA-DFT**) provides consistently strong performance on both layouts, whereas full fine-tuning (**GAMMA-HA-FFT**) provides the strongest performance on the first layout, but weaker performance on the second layout. At first, we hypothesize this is because the second layout is more complex, but the number of human coordination trajectories available for training ($N = 11$) is significantly less than the first layout ($N = 37$). With more diverse potential strategies and less human data, we find the

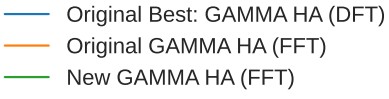

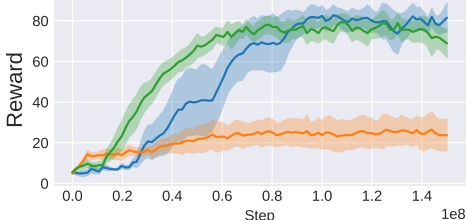

Figure 14: With larger KL Divergence penalty coefficient ($\beta$ in Eq [1]), the new full-model fine-tuning (FFT) largely improves the original FFT and achieves a comparable performance with the original best decoder-only fine-tuning (DFT) method.

data is insufficient to fully fine-tune the generative model for the second layout. Since the entire model is fully fine-tuned, there is a higher chance that the model overfits the training human samples when the amount of human data is extremely small. Therefore, we believe the results of the FFT model could be improved were we to collect more data. However, it is realistic to test the scenario where limited human data is available, since human data can be quite expensive and difficult to collect. We find that in this low data regime, **GAMMA HA DFT** still provides excellent performance. Whether to fine-tune the encoder is a design choice that can be tuned for a particular domain based on the available data and the performance against simulated agents (since Figure 13 shows that FFT performed poorly here as well).

| Agent | Training data source | Counter circuit | Multi-strategy Counter |
|---|---|---|---|
| FCP | FCP-generated population | $32.11 \pm 1.50$ | $44.22 \pm 2.15$ |
| FCP + GAMMA | | $\mathbf{77.75 \pm 2.09}$ | $\mathbf{74.44 \pm 2.12}$ |
| CoMeDi | CoMeDi-generated population | $69.62 \pm 3.31$ | $\mathbf{32.02 \pm 1.89}$ |
| CoMeDi + GAMMA | | $\mathbf{77.77 \pm 2.34}$ | $32.34 \pm 1.79$ |
| PPO + BC | human data | $61.77 \pm 2.53$ | $97.73 \pm 1.90$ |
| PPO + BC + GAMMA | | $72.32 \pm 1.71$ | $95.72 \pm 1.75$ |
| GAMMA HA DFT | human data + FCP-generated population | $82.82 \pm 1.84$ | $\mathbf{103.76 \pm 1.96}$ |
| GAMMA HA FFT | | $\mathbf{91.84 \pm 2.91}$ | $34.05 \pm 2.01$ |

Table 5: Human evaluation results (outdated). Our methods (GAMMA) show significant improvements. However, GAMMA HA FFT is not stable on "Multi-strategy Counter".

## I.2 Large regularization mitigates the problem of insufficient human data

Given the hypothesis that the human dataset is too small compared to the complexity of the *Multi-strategy Counter* environment, we find out that with a larger regularization over the fine-tuned model on the original model, we can mitigate this issue. See Figure 14.

## J Can we condition the Cooperator policy on $z$?

One potential future work to improve the efficiency of online adaptation is to condition the policy on $z$. During testing with novel humans, the Cooperator can then infer the $z$ of the human player and adapt to that $z$.

This method is orthogonal to our contributions where we focus on sampling partners with different $z$ to train the Cooperator. We did some preliminary work to test the $z$-conditioned Cooperator. As shown in Figure 15, the performance of the $z$-conditioned Cooperator is not stable, it learns faster but the performance decreases when it is trained longer. Therefore, we do not include this method in

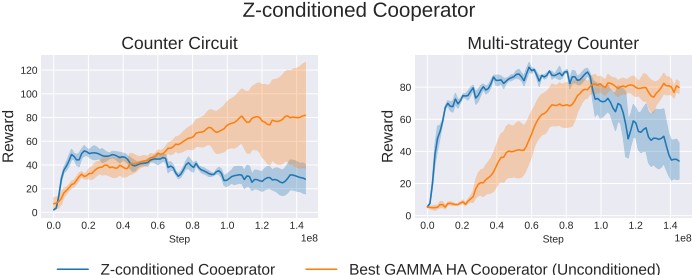

Figure 15: Performance of $z$-conditioned Cooperator. The $z$-conditioned Cooperator reaches a higher reward in the Multi-strategy Counter. The performance decreases after the peak since the $z$-conditioned policy overfits the encoder.

our main experiment, but future work about how to better train the $z-$ conditioned Cooperator is promising.

