# OpenReview forum: "Learning to Cooperate with Humans using Generative Agents"
_NeurIPS.cc/2024/Conference — NeurIPS 2024 poster_

### Official Review · Reviewer_UQLZ · 2024-06-18

**Soundness:** 3
**Presentation:** 4
**Contribution:** 3
**Rating:** 7
**Confidence:** 4

**Summary:**

This work proposes a novel method to train agents to coordinate with humans in team tasks. The main contribution is that instead of training an RL cooperator policy based on a limited number of human models, the authors use a generative model to capture the latent variable representation of human policy. The learned continuous latent space is believed to cover the strategy space of a specific team task, therefore can be used to sample diverse human policies during agent training. Experimental results confirm the benefit of introducing generative models in comparison to previous methods using behavior cloning human models or population-based RL models.
The authors also conduct human evaluations to pair trained agents with novel human participants via crowd-sourcing. The team performance and perceived utilities are higher when human players are paired with agents trained with the proposed method.

**Strengths:**

1. This paper is well written and easy to follow. The introduction and related work sections motivate the work and position it among previous literature well. The problem definitions and methods section are clear. I enjoyed reading it.

2. The problem of learning to cooperate with (unseen) humans is generally underexplored. This work is relatively novel by introducing a generative model and the adaptive sampling method to capture diverse human strategies.

3. The experiments are well designed. The proposed method is evaluated among a variety of task scenarios against traditional methods. Results show that the proposed GAMMA method improves the team performance in comparison to baseline methods such as population RL or behavior cloning. The evaluation environment, i.e. Overcooked-AI, is a rather common benchmark facilitating comparison with other work in the future.

4. The authors conducted evaluations with real human participants, instead of purely relying on simulated human models. The results confirm the actual ad-hoc teamwork performance of the proposed agent with unseen human participants, yielding high ecological validity.

**Weaknesses:**

1. This work lacks a formal analysis about the constructed latent space of human policy given the training data set. There seems to be no theoretical guarantee or quantitative analysis about the generalizability of the proposed generative model over training data, as claimed in lines 198-203. If the collected trajectories in D are biased / do not cover the entire strategy space, the performance of the proposed method might be impacted. Provide more details about Fig. 1, or quantitative analysis of the generalization / interpolation properties would help the readers to value the proposed method.

2. Details about the human data used during training are missing. Are they independently collected from human teams with optimal / consistent team strategies? The setup is very important because the policy distribution of the human data would impact the learned generative models and human proxy models used during training. Explaining the data collection setups and comparing the nature of collected trajectories from agents / humans would enhance the paper.

3. Training a Cooperator policy via RL given a partner policy conditioned on a distribution of latent variables is not trivial. There seems to be a hidden assumption about the distribution parameters to make sure partner policies do not differ too much so the RL policy can converge. The paper does not report details about this process.

4. The hidden assumption of human-adaptive sampling is that human data are closer to the target partner policy compared to synthetic data, so we want to fine-tune the generative model and Cooperator policy on it. This might not be the case if human trajectories are noisy.

5. The authors mention type-based reasoning in the related work section but do not include them in the baselines [1]. It is worth discussing how the proposed method and type-based methods perform differently, given different human policy distributions.

6. One potential future direction is to extend the framework to be online adaptive to the target partner during interaction. For example, using a similar technique to infer the human-centered latent Gaussian distribution of a specific individual human, and change the agent action policy accordingly on the fly.

[1] Zhao, M., Simmons, R., & Admoni, H. (2022, October). Coordination with humans via strategy matching. In 2022 IEEE/RSJ International Conference on Intelligent Robots and Systems (IROS) (pp. 9116-9123). IEEE.

**Questions:**

1. How are different policies sampled and visualized on the latent strategy space in Fig 1?

2. When sampling different latent variables of the partner at each episode, could PPO guarantee to learn an optimal cooperator policy?

3. Are trajectories in the human data set all from the same human-human team with a consistent strategy or a population of individual human players with different strategies randomly paired into teams? Are those human strategies optimal or not?

4. How is the held-out performed on human data and self-play agents respectively? Line 475 only mentions a 30%-70% split without mentioning if agent policy type is stratified.

5. What does this sentence mean in lines 286-288? If the held-out is properly performed on human data, e.g. removing trajectories independently collected from different human participants, PPO-BC has no chance of exploiting information about the unseen evaluation data set. This question might be related to the previous concerns about human data distribution and collection details.

6. How novel are the strategies in human evaluation compared to the original training human data?

7. How accurate is the BC model? What is the action prediction score?

**Limitations:**

In the supplementary materials, the authors acknowledge the limitation of using limited human data which prevents the generative model from learning from scratch. I actually see this as a strength of the paper since human data is usually expensive to collect in practice. While the limited amount of human data is acceptable, the limited characteristics of human data might hurt the further application of the proposed method. As discussed earlier, if the authors could consider examining the quality and features of the used data set, it would shed light on learning to collaborate with humans in ad-hoc teamwork.

There are a few inconsistent patterns within the experimental results (e.g. Figure 5 (a) and Figure 6 (b)) where the proposed method does not achieve better performance than the baselines. The authors attribute those failed cases to limitations of the training data collection process. This actually illustrates the limitations of the proposed method in solving the 2 main challenges claimed in the paper: 1) lack of real (high-quality) human data and 2) the difficulty of synthetically covering the large strategy space of human partners.

---

> ### Author Rebuttal · Authors · 2024-08-07
>
> Thank you for your detailed review and positive feedback on our comprehensive real-human study. We're glad you find it has high ecological validity.
>
> > There are a few experiment results (Fig 5a and 6b) where GAMMA + HA (FFT) is not good on Multi-strategy Counter.
>
> We acknowledge that the FFT method originally performed poorly due to limited human data. Our further experiments found that increasing the regularization can mitigate the problem of lack of human data, enabling GAMMA HA FFT to provide good performance in Multi-Strategy Counter as well. See global response and Fig (e) in the rebuttal PDF for details.
>
> > One potential future direction is to infer the human latent variable […]
>
> Thank you for pointing out this! We agree that we can also make $z$ part of $\pi$. **Note sampling from $z_i$ and making $z_i$ part of $\pi$ are orthogonal ideas and can be combined to train the cooperator.**
>
> We implement this idea by allowing the Cooperator to condition on $z$ inferred by the VAE. We present the results in Fig. (c) of the rebuttal PDF and find that this improves both learning efficiency and performance on Multi-strategy Counter.
>
> > Are trajectories in the human dataset all from the same human-human team […]
>
> For layouts proposed by [1], we use their open-source human from randomly paired human-human teams. For the new Multi-strategy Counter, we collect data similarly, sampling players randomly and explaining the game rules before collecting their gameplay data. These strategies are not always optimal; in Figure (f) of the rebuttal PDF, 49.5% of the players are novice overcooked players. People played according to their own strategies without prompts. We will add this information to the paper.
>
> > Details of Fig 1.
>
> Please refer to our common response.
>
> > Could PPO guarantee to learn an optimal cooperator policy?
>
> In general, gradient-based RL is not guaranteed to converge in multi-agent settings [3].  Although we can not ``guarantee’’ PPO to learn optimally, we give it enough training time (1.5e8 steps) to converge.
>
> > There seems to be no theoretical guarantee [...]
>
> We will soften the claims in lines 198-203. We acknowledge assuming the human data for training is representative of the target population during the human study. We will add the following limitation to the paper:
>
> "We assume the human dataset for VAE training and human-adaptive sampling matches the distribution of human players during evaluation. This may not hold if the dataset quality is low. Addressing distribution shift from training to testing is important and will be our future work."
>
> > How is the held-out performed on human data and self-play agents respectively? [...]
>
> The 70%-30% split is only used to determine the validation dataset for VAE training. The held-out agents will never be used for training VAEs or Cooperators.
>
> > [...] examining the quality and features of the used data set, [...]
>
> We provide a detailed introduction to human data here and will add this to the paper. For the human dataset in the original Overcooked paper [1], their open-sourced dataset contains ``16 joint human-human trajectories for Cramped Room environment, 17 for Asymmetric Advantages, …, and 15 for Counter Circuit.’’ with length of $T\approx 1200$. In Multi-strategy Counter, we collect 38 trajectories with length $T\approx 400$ which is closer to the actual episode length during training.
>
> For the human study, we have made additional plots that show the participants’ distribution of experience levels for playing Overcooked. In Figure (f) in the rebuttal PDF, 49.5% of players agreed that they had no experience of playing Overcooked. We also recommend looking at the demos on our website, where we show videos of humans using different strategies.
>
> >  What does this sentence mean in lines 286-288? [...]
>
> While the proxy metrics are relatively cheaper to run than real human evaluations, both our work and [1] find that they favor the PPO-BC method, and thus can be misleading for final coordination performance with real humans. **Again, we emphasize that the most important experiment is the human evaluation (Figure 6) and GAMMA is statistically significantly better.**
>
> > How novel are the strategies in human evaluation […]?
>
> For all human players, we only give them basic instructions about the game, so they are free to use unique strategies. Especially for Counter Circuit, we recruit an entirely new population of participants using a different platform and potentially different recruiting criteria of [1]. Therefore, this test-time population could be quite different from the training dataset [1] released several years ago.
>
> > How accurate is the BC model? [...]
>
> The accuracy is around 55-60%. Under self-play, our BC model reaches 40-80 scores (2-4 soups). We use the same dataset [1] to train the BC model as prior work.
>
> > Compare GAMMA with type-based reasoning work [2].
>
> We thank the reviewer for pointing us to [2], which has a similar setting to our PPO + BC + GAMMA method. During training, [2] clusters human trajectories into different strategies and learns the best response for each, while GAMMA uses a VAE over the entire human dataset to train a universal Cooperator. When facing a novel human, [2] infers the human's strategy and applies the best response to it, whereas GAMMA uses the universal Cooperator. Thus, while [2] is limited to partner strategies in the human dataset, PPO + BC + GAMMA can potentially generate novel partner policies through interpolation. Moreover, unlike [2], which only uses human data, GAMMA can combine synthetic and human data with controllable sampling like GAMMA HA. We will cite this work and include the discussion in the paper.
>
>
> [1] M. Carroll, et al. On the utility of learning about humans for human-ai coordination.
>
> [2] Zhao, et al. Coordination with humans via strategy matching.
>
> [3] Mazumdar, et al. "On finding local nash equilibria (and only local nash equilibria) in zero-sum games."

---

> > ### Comment · Reviewer_UQLZ · 2024-08-12
> >
> > Thank you for your rebuttal. It resolves some of my concerns and helps me better understand the experiment details. However, my concerns about formal analysis of latent space and the learning process of the cooperator policy remain. I will keep my original rating.

---

### Official Review · Reviewer_VM8E · 2024-06-20

**Soundness:** 3
**Presentation:** 4
**Contribution:** 3
**Rating:** 4
**Confidence:** 5

**Summary:**

This paper proposes GAMMA, a generative model that learns a latent representing partner agents' strategy. Through interpolation, the latent can be used generate diverse partners to train ego agents for human-AI cooperation. Authors studied training the generative model with both simulated data and real-human data and conducted evaluation with both human proxies and real humans. Experiment results show GAMMA improves both objective performance and human subjective ratings.

**Strengths:**

1. This paper is well-written and easy to comprehend.

2. The targeted issue of diversity of partner agents is important in zero-shot coordination and human-AI coordination.

3. Human study is conducted at relatively large scale to show the efficacy of GAMMA in human-AI cooperation.

**Weaknesses:**

1. The 'state-of-the-art' baselines in this paper are outdated (especially FCP). I would be happy to see comparison against works like [1],[2],[3], which are also targeted at zero-shot coordination.

  [1] Mahlau, Yannik, Frederik Schubert, and Bodo Rosenhahn. "Mastering Zero-Shot Interactions in Cooperative and Competitive Simultaneous Games." arXiv preprint arXiv:2402.03136 (2024).

  [2] Yan, Xue, et al. "An Efficient End-to-End Training Approach for Zero-Shot Human-AI Coordination." Advances in Neural Information Processing Systems 36 (2024).

  [3] Zhao, Rui, et al. "Maximum entropy population-based training for zero-shot human-ai coordination." Proceedings of the AAAI Conference on Artificial Intelligence. Vol. 37. No. 5. 2023.

2. In Fig. 4,5, GAMMA's improvements over baselines seem marginal.

3. The interpolation visulization needs better explanation. What do the points in Figure 1 stand for? Is it policy $\pi$ or hidden vector $z$? I can't find the reletive information to comprehend this figure.

4. Adding to the last question, utilizing interpolation to enhance partner diversity is not new in zero-shot coordination field. PECAN [4] directly interpolates parnter policies through policy ensemble instead of hidden vector $z$ (see figure 4 in [4]). What is the advantage of GAMMA by interpolating at the level of hidden vectors? Does it offer better partner diversity? This should be at least discussed in the paper.

[4] Lou, Xingzhou, et al. "Pecan: Leveraging policy ensemble for context-aware zero-shot human-ai coordination." arXiv preprint arXiv:2301.06387 (2023).

5. Typos:
"data-hungry seuential decision making algorithms...", on page 1

"We propose a novel, ecnonomical way" on page 2

e.t.c.

**Questions:**

See weaknesses.

Please at least add MEP [3]  to the baselines in your experiments (since it has very similar pipeline as FCP in your paper and easy to run). I would happily increase my score upon seeing such comparisons.

[3] Zhao, Rui, et al. "Maximum entropy population-based training for zero-shot human-ai coordination." Proceedings of the AAAI Conference on Artificial Intelligence. Vol. 37. No. 5. 2023.

**Limitations:**

Limiations are discussed in the appendix.

---

> ### Author Rebuttal · Authors · 2024-08-07
>
> Thank you for your review! For typos, we will correct them in the next version. We address your concerns below.
>
> >  The baselines are outdated.
>
> Thanks for the suggestion, we have run MEP (see Fig 2 in the rebuttal PDF) and show that GAMMA can still improve its performance.
>
> We agree all the methods (MEP, E3T, SBRLE, and PECAN) that you mentioned are important and newer works in the human-AI cooperation field. We would like to point out that our GAMMA can not only 1) improve the diversity of the population with generative agents, which those prior works did; but also 2) combine another line of work of human modeling from human data under our human-adaptive sampling techniques.
>
> To test 1), we need to show that GAMMA can improve diversity by generative agents trained on any synthetic population. Since FCP is the most popular and CoMeDi is one of the latest works, we believe these are two representative populations to test GAMMA. We will include the additional results with MEP in the revised version of the paper.
>
> > GAMMA's improvements seem marginal in Fig. 4,5
>
> - First of all, for the most important human experiment in Fig 6, it is statistically significant that GAMMA is better than baselines (p < 0.05).  See more details in the common response.
>
> - Regarding Cramped Room and Forced Coordination in Figure 4, we agree GAMMA does not provide significant improvement. The main reason is that the environments are so simple that the optimal policies can be easily found for certain algorithms. In more detail, in Cramped Room, the agents simply need to fill the pot whenever it is available. In Forced Coordination, the left agent only needs to place the onions and dishes on the central table, while the right agent only needs to collect the onions for the pot and use the dish to complete a soup order. Note that for more complex environments like Coordination Ring or Counter Circuit, GAMMA gives significant gains.
>
> - Regarding Figure 5, we choose these experiments to follow existing works [1] [4]. We agree that GAMMA’s performance is only comparable to the strongest baseline but not significantly better. **On the other hand, we note there is a problem with this experiment.**  As shown in [5], the performance evaluated by the human proxy model (a BC model) is not predictive of performance with real humans since BC models are trained on limited data, are not reactive or adaptive in the same way as humans and therefore have considerably different behavior than real human agents on evaluation. While the proxy metrics are relatively cheaper to run than real human evaluations, both our work and [5] find that they can be misleading in terms of final coordination performance with real humans. **Again, we emphasize that the most important experiment is the human evaluation (Figure 6) and GAMMA is statistically significantly better.**
>
>
>
> > What do the points in Figure 1 stand for?
>
> We provide a detailed response in the common rebuttal. It is the latent vector $z$. For all population $P$, each point of $P$ is computed by 1) first sampling an agent in $P$; 2) using the agent to generate an episode with self-play, and 3) encoding the episode into $z$ using VAE. These high-dimensional latent vectors are compressed to 2D using t-SNE. We promise to add this explanation to the paper in the future.
>
> > Compared to PECAN [2], what is the advantage of GAMMA by interpolating at the level of hidden vectors?
>
> Thank you for pointing us to the PECAN paper [2], we will include a citation of this work and the following discussion of the advantages of using GAMMA in the revised version of the paper:
>
> ''PECAN [18] proposes an ensemble method to increase the partner population diversity. In PECAN, partner agents are categorized into three groups based on their skill levels. With GAMMA, the skill levels are automatically learned and encoded in the latent variable $z$. And by sampling different $z$, generative partner agents with different skill levels are thus sampled.
>
> GAMMA offers several advantages over this method. First, GAMMA provides a controllable way to sample only from a target subpopulation (e.g., humans) by adaptive sampling over the latent space. Second, it is known that with interpolation on the latent variable $z$ [3], generative models can produce novel samples beyond the weighted sum of low-level pixels/actions. ''
>
>
> > Typos: "data-hungry seuential decision making algorithms...", on page 1.
>
> Thank you for your time and attention to detail. We promise to fix that in the future version.
>
>
> [1] C. Yu, J. Gao, W. Liu, B. Xu, H. Tang, J. Yang, Y. Wang, and Y. Wu. Learning zero-shot cooperation with humans, assuming humans are biased.
>
> [2] Lou, Xingzhou, et al. "Pecan: Leveraging policy ensemble for context-aware zero-shot human-ai coordination."
>
> [3] Radford, Alec, Luke Metz, and Soumith Chintala. "Unsupervised representation learning with deep convolutional generative adversarial networks."
>
> [4] D. Strouse, K. McKee, M. Botvinick, E. Hughes, and R. Everett. Collaborating with humans without human data.
>
> [5] M. Carroll, R. Shah, M. K. Ho, T. L. Griffiths, S. A. Seshia, P. Abbeel, and A. D. Dragan. On the utility of learning about humans for human-ai coordination.

---

### Official Review · Reviewer_pvTn · 2024-07-03

**Soundness:** 2
**Presentation:** 4
**Contribution:** 3
**Rating:** 4
**Confidence:** 5

**Summary:**

The paper addresses the challenge of training AI agents that can effectively coordinate with diverse human partners in cooperative tasks without prior interaction. This work demonstrates that using generative models to create diverse training partners can significantly improve an AI agent's ability to coordinate with novel human partners in cooperative tasks. The proposed approach offers a promising direction for developing more adaptable and effective AI assistants for human-AI collaboration scenarios.​​​​​​​​​​​​​​​​

The key contributions of the work are listed as follows:

1. The work proposes a novel approach, Generative Agent Modelling for Multi-agent Adaptation (GAMMA), that uses generative models to produce diverse partner agents for training a robust Cooperator agent. Through conducted experiments using the Overcooked environment, the authors demonstrate that GAMMA consistently outperforms baselines in zero-shot coordination with novel human partners, both in terms of task completion and subjective human ratings.

2. The authors show the flexible data integration of GAMMA, i.e. it can be trained on both simulated agent data and real human interaction data. This addresses the dual challenges of limited human data and insufficient coverage of human behaviours in synthetic data.

3. The authors also propose a method, Human-Adaptive Sampling, to efficiently incorporate a small amount of human data to bias the generative model towards producing more human-like partners during training.

4. The study brings together and directly compares two previously separate approaches to zero-shot coordination: training on simulated population data, and training on human data.

**Strengths:**

1. *The research problem is of great significance*:
This work studies zero-shot human-AI coordination, which is a key topic for applying AI in the real-world. The key bottleneck the work aims to address is the coverage of behaviours by the simulated humans obtained by behaviour cloning over datasets of human cooperation, or by using MARL. This work offers a promising direction for developing more adaptable and effective AI assistants for human-AI collaboration scenarios.​​​​​​​​​​​​​​​​

2. *The paper is well-written and well-organised*:
Overall, the paper is well-written, and very-well-organised. Despite the issues mentioned in the weaknesses section below, the motivation, the proposed method, and all experiment results are clearly delivered. Overall, I quite enjoy reading the paper.

3. *The conducted experiments are thorough and the hypotheses are clearly formulated*:
It's pleasant to see that the hypotheses to verify are clearly listed at the beginning of Section 5, and the following results are also demonstrated following the same order of hypotheses.

4. *The proposed method is very intuitive*:
In this context, the term "intuitive" is used to convey a positive quality. It makes a lot of sense to me that human partners can be represented by latent variables, like the user embeddings used in recommendation systems. It also makes sense that depending on the user embeddings leads to higher coverage of the behaviours of simulated human partners.

**Weaknesses:**

1. *Results lack of statistical significance, thus persuasiveness*:
Despite all the good aspects of the work, including the motivation and the intuition, my biggest concern is that the improvement of the performance doesn't look significant. First, in the Cramped Room in Figure 4, there seems to be overfitting after $5e7$ steps. In the meantime, GAMMA doesn't lead to improvement on FCP at that time step. Similarly, there seems no (statistically sifnificant) difference between FCP and CoMeDi with and without GAMMA at the $5e7$-th step in Forced Coordination. Considering that the performance of both FCP and CoMeDi decrease after $5e7$ steps in Forced Coordination as well, a more rigorous conclusion could be that GAMMA can prevent the overfitting, instead of improve the performance. Secondly, the performance shown in Figure 5 also suggests that GAMMA doesn't improve the performance, since the orange curves (PPO + BC + GAMMA) twist with the blue curves (PPO + BC). Even though the methodology is convincing, the results make me doubt whether the proposed GAMMA can indeed lead to significant improvement.

2. *Need to illustrate the necessity of sampling from $z_i$*:
In Section 4.2, the authors illustrate the human behaviour representation $z_i$ is used to generate batches of MDPs. A straightforward and intuitive alternative practice is simply to make $z_i$ a part of conditionals for $\pi$. The authors need to provide either a rational in more details or more empirical evidence on the chosen implementation, i.e. using $z_i$ to generate $\\{M_{z_i} \\}_{i=1}^{N}$.

3. *Length of games during evaluation*:
As an experienced player of Overcooked, the 60-second setup used in human evaluation looks too short to me. According to my own experience, for human-human cooperation, it usually takes a whole trial of the game, i.e. >120 seconds. This makes me doubt whether improvement by GAMMA is only on the adaptation efficiency. Moreover, this also makes me doubt whether the improvement observed in the evaluation is only due to some randomness. I would be keen in seeing more trials of each method with human partners in longer games.

**Questions:**

1. Why is "training on human data is highly effective in our human evaluation" counterintuitive?
The authors mention in line 95 that "contrary to past published results, training on human data is highly effective in our human evaluation". I got confused by this claim as it sounds intuitive to observe better performance by using human data in human evaluation.

2. Can increasing sampling temperature also lead to greater coverage over the strategy space?
In line 200, the authors claim that "the generative model has significantly greater coverage over the strategy space". A natural alternative could be simply increasing the sampling temperature. So, I wonder if this simple trick could also boost the performance/efficiency of the same baseline methods?

3. Is H3 a hypothesis?
From my perspective, the H3 stated from line 261 to line 263 is not a hypothesis.

4. Why is it useful to evaluate the method by a population of held-out self-play agents?
As stated in line 289, the human evaluation is completed on "a population of held-out self-play agents". After reading the paragraph from line 282 to line 289, I'm still unclear about the rationale of doing so. More importantly, line 95 state that human data is not so effective in human evaluation. Then, why is fine-tuning on human data a good evaluation?

5. How are the number of policies (8) and the numbers of random seeds (5) decided in human evaluation?

**Limitations:**

Please refer to the weaknesses section.

---

> ### Author Rebuttal · Authors · 2024-08-07
>
> We thank you for your detailed review. We address your questions below.
>
> > Results lack statistical significance
>
> - First of all, for the most important human experiment, GAMMA is statistically significantly better (p < 0.05).  See the common response in detail.
> - For Cramped Room and Forced Coordination in Fig 4, we agree GAMMA does not provide significant improvement. The main reason is that the environments are so simple that the optimal policies can be easily found. Specifically, in Cramped Room, the agents simply need to fill the pot whenever it is available. In Forced Coordination, the left agent only needs to place the onions and dishes on the central table, while the right agent only needs to collect the onions for the pot and use the dish to complete an order. Note that for more complex Coordination Ring or Counter Circuit, GAMMA gives significant gains.
> - Regarding Fig 5, we choose these experiments following [2] [4]. We agree that GAMMA’s performance is only comparable to the strongest baseline. **Note there is a problem with this evaluation.**  As shown in [1], the performance evaluated by the BC human proxy model is not predictive of human study. The BC model trained on limited data, is not reactive or adaptive in the same way as humans and therefore has considerably different behavior. While the proxy metrics are relatively cheaper to run than real human evaluations, both our work and [1] find that they can be misleading for performance against real humans. **Again, we emphasize that the most important experiment is the human evaluation (Fig 6) and GAMMA is statistically significantly better.**
>
> > Illustrate the necessity of sampling from $z_i$
>
> We first explain the rationale of sampling from $z_i$. Our primary idea is to build a population and train a cooperator against it. We choose VAE to represent this distribution and here $z_i$ serves as the latent vector to generate partner policies.
>
> We agree that we can also make $z_i$ part of $\pi$. **Note that sampling from $z_i$ and making $z_i$ part of $\pi$ are orthogonal ideas and can be combined together.** We have conducted two additional experiments.
>
> First, we make the Cooperator condition on the $z$ inferred by the VAE. In Fig (c) in the rebuttal PDF, we found this approach improves both learning efficiency and performance on Multi-strategy Counter.
>
> Second, as another baseline, we use the $z$-conditioned decoder of the VAE as the policy. The results are presented in Fig (d) in the rebuttal PDF. We find that the performance of the $z$-conditioned decoder is not good compared to the Cooperators trained by sampling partners from the VAE.
>
> > The length of games is short
>
> We appreciate your insight as an experienced Overcooked player. We chose this time limit to be consistent with prior works; 60s for FCP and 40s for CoMeDi. In our evaluation, the human-AI team can usually finish 2-5 soups (60-100 scores) in 60s.
>
> > Can increasing the sampling temperature also boost the performance?
>
> Making the partner's actions more random could simulate a less skilled agent. Our VAE training dataset already includes random agents, enabling it to interpolate between expert and random agents.
>
> The VAE offers a significant advantage over merely adding noise to players' actions, as it can combine different skillful strategies. For example, if partners follow strategies A, B, or C, adding noise to A produces an inexpert A. In contrast, the VAE can create a partner proficient in all strategies, like ACBAAC. We hypothesize the VAE better covers possible skilled partner strategies (Fig 1), and our experimental results support this hypothesis.
>
>
> > Why mention ``training on human data is highly effective’’ particularly in line 95?
>
> We agree that it is intuitive that adding human data should improve performance. Prior work [2] shows that FCP > PPO-BC in four layouts including our Counter Circuit, and PPO-BC > FCP for Asymmetric Advantage. In our experiments, we found PPO-BC also effective for both Counter Circuit and Multi-strategy Counter. We hope our findings can encourage the community to engage in further discussion and empirical studies on this problem.
>
> > Why use held-out self-play agents in Fig 5b? Line 95 states that human data is not so effective in human evaluation. Then, why is fine-tuning on human data is good?
>
> We use held-out self-play agents to follow existing works [2, 4]. The reason to use self-play agents is that a human-proxy model is not a good evaluation metric, as shown in prior work [1]. **We emphasize again that the most important experiment is the human evaluation (Fig 6) and GAMMA is statistically significantly better.**
>
> Our finding shows that human data is quite effective, while prior work [2] mentioned in line 95 focuses on training without human data. We hope our findings can encourage more attention to this problem.
>
> >  How are the number of policies (8) and the number of random seeds (5) decided in human evaluation?
>
> We follow the same number of five random seeds as prior works [2, 3] to ensure the statistical significance of our results We did two separate human evaluation rounds of the two layouts we used. For each round, only one layout would be tested. We first shuffled the order of eight algorithms randomly, and then for each algorithm, we sampled one of five random seeds to play against humans.
>
> > H3 is not a hypothesis.
>
> Thanks. We will revise H3 to, “Generalizing to novel humans: we hypothesize that GAMMA will generalize more effectively to novel humans than prior methods, showing improved performance in both game score and human ratings.”
>
> [1] M. Carroll, et al. On the utility of learning about humans for human-ai coordination.
>
> [2] D. Strouse, et al. Collaborating with humans without human data.
>
> [3] R. Zhao, et al. Maximum entropy population-based training for zero-shot human-ai coordination.
>
> [4] C. Yu, et al. Learning zero-shot cooperation with humans, assuming humans are biased.

---

### Official Review · Reviewer_igMC · 2024-07-10

**Soundness:** 3
**Presentation:** 3
**Contribution:** 3
**Rating:** 6
**Confidence:** 4

**Summary:**

The authors propose a novel method for training agents that effectively cooperate with humans by leveraging generative models that can sample the space of agent behaviors

**Strengths:**

- Well-written and easy to follow
- Experimental setup is clear, well-documented, and expansive
- Code and even a live demo are available (very cute demo by the way!)
- Novel method in an important and growing area

**Weaknesses:**

- Experimental results are okay, but not clear that GAMMA consistently outperforms baselines. This is true in both the simpler and more complex settings.  Even in cases where GAMMA-based methods have higher performance, the confidence intervals seem to overlap making it difficult to say whether results are simply due to chance. See Questions below for a few more things I'm wondering about regarding results and significance. Happy to raise my score if authors can show that the proposed method fairly consistently outperforms baselines in a statistically significant way (either by correcting me if I'm wrong in my interpretation of current results, or by increasing sample size to decrease confidence intervals).

Minor suggestions (did not affect score):
 - increase text size in fig 2
 - report results in a table (can go in the appendix) as it is hard to tell from the graphs alone whether certain methods outperform others in statistically significant ways

**Questions:**

- how well would this approach generalize to more realistic settings? It seems like moving to the more complex setting led to reduced gains compared to baselines.
- relatedly, how complex are the strategies learned by the Cooperator? Can they deal with humans who learn, adapt, and change their own strategies?
- can the Cooperator instead be meta-learned and adapt online?
- not entirely clear what is meant by "Error bars [4] use the Standard Error of the Mean (SE) for statistical significance (p < 0.05)". Are you saying that error bars correspond to 95% CI? or that they correspond to standard error?
- following up on that, which of the results are statistically significant? hard to tell from the figures.

**Limitations:**

- Authors note human diversity is critical, but it's somewhat unclear whether human data comes only from WEIRD populations (almost certainly yes since participants were recruited from Prolific). This is worth noting in the human eval/limitations/impact sections since results may not be representative of broader populations (especially since authors mention assistance for elderly or disabled people as potential future application). A potential social impact of this is that a) agents may best serve only the demographics they were trained on, and b) agents may lead to reduction in creativity and diversity of strategies by forcing humans to adapt to the limited range of strategies that the agents are familiar with.
- Authors do mention wanting to train on larger datasets in limitations, but also worth mentioning that current experiments are in very constrained environments and only in 2-player settings. In addition to dataset size, scaling along both of these dimensions would be important future work.
- If I understand correctly, the human data used for fine-tuning or for creating the generators is collected either in solo-play or interaction with a simple existing policy. It seems unlikely then, that the GAMMA-based methods could deal with humans learning, adapting, and changing their strategies. Perhaps worth mentioning this as well.

---

> ### Author Rebuttal · Authors · 2024-08-07
>
> We thank the reviewer for your detailed and insightful feedback. We are grateful that you took the time to check out our demo. We hope the following responses address your concerns.
>
> > Does the proposed method GAMMA fairly consistently outperform baselines statistically significantly?
>
> GAMMA does outperform baselines, which we clarify in the global response as well. The human evaluation results (Fig 6 and 7) should be considered the gold standard evaluation, and here GAMMA shows a statistically significant improvement over all baselines on both layouts. We provide a table of human evaluation results (Table 1) in our rebuttal PDF which makes this more clear. Other simulated metrics (Fig 4 and 5) are not always predictive of human evaluation performance.
>
>
> > What is the error bar in Fig 6? Which of the results are statistically significant?
>
> Because the error bars correspond to the Standard Error of the Mean (SE), if two error bars are non-overlapping, the results are considered statistically significantly different at the p < 0.05 level. For more information on this, please see: G. Cumming, F. Fidler, and D. L. Vaux. Error bars in experimental biology. The Journal of cell 377 biology, 177(1):7–11, 2007. Table 1 in the rebuttal PDF bolds the GAMMA models that are significantly better than the baseline model trained with the same data.
>
> > How complex are the strategies that the Cooperator can learn? Can they deal with adaptive humans?
>
> In our evaluation with real humans, participants are free to change the strategy they play during the episode. In fact, since many of our participants are novice players who have little experience with the game Overcooked or with video games in general, often their performance continues to improve as they play. We show in Figure (e) in the rebuttal PDF that performance improves consistently with the number of trials, providing evidence that humans are changing their gameplay throughout the evaluation. Since our method provides the best performance when playing with real humans, we believe it is best at adapting to their changing game play.
>
> > How is the human training data collected? If I understand correctly, the human data is collected either in solo play or interaction with a simple existing policy. It seems unlikely then, that the GAMMA-based methods could deal with humans learning, adapting, and changing their strategies.
>
> We would like to clarify that the data is collected by randomly assigning two human players to play together, so it consists of human-human trajectories where players can adapt and change their strategies. Further, as we show above, in our evaluation human players do learn to improve their performance over time, showing evidence of learning and adapting throughout the evaluation.
>
> > The gains are reduced for more complex settings.
>
> We believe GAMMA is a general approach that can still be applied to more complex settings. Figure 6 shows that GAMMA HA DFT still provides statistically significantly higher performance than competitive baselines in the more complex layout. In the original experiment, FFT is less effective on Multi-strategy Counter due to the relative lack of human data. In Figure (b) in the rebuttal PDF, We solve this issue by using a larger KL Divergence penalty coefficient and now both FFT and DFT achieve the best performance on Multi-strategy Counter.
>
> > Can the Cooperator instead be meta-learned and adapt online?
>
> Currently, our Cooperator adapts according to the history of the other agents. Thank you for the suggestion. We agree that we can also make $z_i$ part of $\pi$. **Here we note, sampling from $z_i$ and making $z_i$ part of $\pi$ are orthogonal ideas and can be combined to train the cooperator.** We have conducted the following experiment.
>
> We implement this idea by allowing the Cooperator to condition on the z inferred by the VAE. We run this experiment and present the results in Fig. (c) of the rebuttal PDF. Our findings indicate that this approach improves both learning efficiency and performance on the Multi-strategy Counter layout.
>
>
>
> > Limitations. The human data in this work may not be representative. The current experiments are only in two-player settings.
>
> Thank you for pointing out these limitations. We will add a limitations section to the revised version of the paper with the following text:
>
> ''In this work, our human studies recruit participants from Prolific, which may not be representative of broader populations. Additionally, our human dataset is limited, which could reduce the diversity of strategies and force participants to adapt to strategies that the Cooperators are already familiar with.
>
> We focus on the two-player setting in this study following prior work [1] [2] because it is a first step toward enabling an AI assistant that could help a human with a particular task. Scaling up to more agents would exponentially increase the dataset size with our current techniques. Therefore, better sampling techniques are needed to address this issue.''
>
>
> [1] D. Strouse, K. McKee, M. Botvinick, E. Hughes, and R. Everett. Collaborating with humans without human data.
>
> [2] C. Yu, J. Gao, W. Liu, B. Xu, H. Tang, J. Yang, Y. Wang, and Y. Wu. Learning zero-shot cooperation with humans, assuming humans are biased.

---

> ### Comment · Reviewer_igMC · 2024-08-07
> **Response to rebuttal**
>
> Thank you for the rebuttal!
> Some of my concerns have now been addressed, but the most major one still remains:
> With regard to standard errors, a gap of one standard error **between the edges of the standard error bars** corresponds to p=0.05. A gap of one standard error **between the means** does **not** correspond to p=0.05. See Fig 5 of the Cumming et al. (2007) paper for an example of this. It is still unclear which results are statistically significant. Consider applying a traditional hypothesis test to get the exact p-values rather than estimating them based on the gap between SEs.

---

> > ### Author Response · Authors · 2024-08-09
> > **Exact p-values**
> >
> > We appreciate your response to our rebuttal. In order to address your concerns, we provide the exact p-value for each comparison between methods for both layouts. In summary, our method GAMMA is statistically significant (p < 0.05) for the majority of comparisons between methods for both layouts, when comparing based on population generation method, and when comparing our best-performing GAMMA model across all baselines per layout. See the following tables for the exact p-values using a t-test. Our human study data has $n=80$ participants.
> >
> > ### Comparison based on data source:
> > | **Comparison**            | **Counter Circuit** | **Multi Strategy Counter Circuit** |
> > |---------------------------|-------------------------------|----------------------------------------------|
> > | **FCP + GAMMA > FCP**      | **6.48e-5**         | **9.99e-22**       |
> > | **CoMeDi + GAMMA > CoMeDi**| **0.0427**        |  0.902     |
> > | **GAMMA HA DFT/FFT > PPO + BC**| **4.08e-14**    | **0.030**                                            |
> >
> > ### Best-performing GAMMA:
> > | **Comparison**              | **Counter Circuit** | **Multi Strategy Counter Circuit** |
> > |-----------------------------|-------------------------------|----------------------------------------------|
> > | **GAMMA HA DFT/FFT > PPO + BC**  |  **4.08e-14**       | **0.030**   |
> > | **GAMMA HA DFT/FFT > FCP**       | **2.04e-57**    | **3.79e-71**         |
> > | **GAMMA HA DFT/FFT > CoMeDi**    |  **7.14e-07**  | **2.18e-105**       |

---

> > > ### Comment · Reviewer_igMC · 2024-08-13
> > >
> > > Thank you, I appreciate this additional analysis. However, it is not correct to pick the best of multiple models and compute significance. This breaks the iid assumption. At a minimum, you would need to apply a multiple comparison correction. Furthermore, you haven't included the comparison of PPO+BC+Gamma against PPO+BC. As a result, it's still not clear whether GAMMA actually improves performance in many of the described settings.

---

> > > > ### Author Response · Authors · 2024-08-14
> > > > **statistical significance test with correction**
> > > >
> > > > We thank you again for your response to our rebuttal. We provide the p-values with Holm-Bonferroni correction below, with the aim of reducing the family-wise error rate. We also include an additional comparison of PPO + BC + GAMMA against PPO + BC as requested. The p-value is compared to 0.0028 as there are n=18 comparisons, thus p=0.05/18=0.0028. Please let us know if you have any additional comments or clarifications.
> > > >
> > > > ### Comparison of best-performing GAMMA against baselines with Holm-Bonferroni correction:
> > > > | **Comparison**            | **Counter Circuit** | **Multi Strategy Counter Circuit** |
> > > > |---------------------------|-------------------------------|----------------------------------------------|
> > > > | **GAMMA HA DFT/FFT > FCP**      | **1.09e-57**          | **7.58e-71**       |
> > > > | **GAMMA HA DFT/FFT > CoMeDi**| **1.23e-06**          |  **2.18e-105**     |
> > > > | **GAMMA HA DFT/FFT > PPO + BC**  |  **8.01e-14**       | 0.12   |
> > > >
> > > > ### Comparison based on data source with Holm-Bonferroni correction:
> > > > | **Comparison**            | **Counter Circuit** | **Multi Strategy Counter Circuit** |
> > > > |---------------------------|-------------------------------|----------------------------------------------|
> > > > | **PPO + BC + GAMMA > PPO + BC** | 0.0031                   | 2.18                         |
> > > > | **FCP + GAMMA > FCP**                    | **1.30e-53**          | **3.00e-21**
> > > > | **CoMeDi + GAMMA > CoMeDi**         | 0.20                      | 5.41

---

### Author Rebuttal · Authors · 2024-08-07

We thank all the reviewers for their detailed reviews and feedback.

To summarize the positive feedback, reviewers recognize the importance of the problem we studied and find the paper well-written with well-designed experiments. We are thankful to reviewer igMC for taking the time to explore the demo. We also appreciate that reviewers find our method intuitive (pvTn) and novel (igMV), and commend our large-scale human study (VM8E) and the high ecological validity (UQLZ) of our work.

We first address a common question asked by all reviewers.

> Which results are statistically significant showing GAMMA is better than baselines?

All reviewers ask whether GAMMA is better than baselines in a statistically significant way. Yes, GAMMA is. Below we provide the results in tabular format as well as an additional explanation to make this more clear.

The real human experiment (the most important evaluation metric for zero-shot human-AI cooperation) is shown in Figure 6. For our error bars, we use the **standard error of the mean (SE), which is calculated as $\text{SE} = \frac{ \text{std dev} }{ \sqrt{n} }$**. We use a total of $n=80$ participants recruited online. Figure 6 in the original draft and the Table below shows we have 1) FCP + GAMMA > FCP; 2) CoMeDi + GAMMA > CoMeDi; 3) GAMMA FFT/DFT > PPO-BC. **Furthermore, for these comparisons 1), 2), 3), the error bars do not overlap, and thus they are statistically significant (p<0.05).** That is, for the same training source, GAMMA consistently improves performance. These results demonstrate that when GAMMA is applied to an existing population method like FCP or CoMeDi it consistently improves their performance, and it can also make better use of human data than the best existing baseline designed for use with human data (PPO-BC). We also note that GAMMA is also consistently rated higher according to subjective human ratings (Fig 7).

| Agent           | Training data source                  | Counter circuit    | Multi-strategy counter  |
|-----------------|---------------------------------------|--------------------|-------------------------|
| FCP             | FCP-generated population | 32.11 ± 1.50       | 44.22 ± 2.15            |
| FCP + GAMMA     |                                       | **77.75 ± 2.09**   | **74.44 ± 2.12**        |
|-----------------|---------------------------------------|--------------------|-------------------------|
| CoMeDi          |CoMeDi-generated population | 69.62 ± 3.31       | **32.02 ± 1.89**        |
| CoMeDi + GAMMA  |                                       | **77.77 ± 2.34**   | **32.34 ± 1.79**        |
|-----------------|---------------------------------------|--------------------|-------------------------|
| PPO + BC        | human data          | 61.77 ± 2.53       | 97.73 ± 1.90            |
| PPO + BC + GAMMA|                                       | 72.32 ± 1.71       | 95.72 ± 1.75            |
|-----------------|---------------------------------------|--------------------|-------------------------|
| GAMMA HA DFT    | human data + FCP-generated population | 82.82 ± 1.84       | **103.76 ± 1.96**       |
| GAMMA HA FFT    |                                       | **91.84 ± 2.91**   | 34.05 ± 2.01            |




We also provide the results of 4 new experiments in the rebuttal PDF, as well as additional information about the population of human participants in our study, which we believe will help address the concerns raised in the reviews.


Next, we provide more details of the new experiments.

> New results for an additional baseline MEP.

As requested by reviewer VM8E, we add MEP as a new baseline. See Fig (a) in the rebuttal PDF. We find GAMMA can improve the performance of MEP.

> New results for GAMMA FFT HA on Multi-strategy Counter.

To answer the question of reviewer UQLZ, we improve GAMMA FFT HA and run new experiments. See Fig (b) in the rebuttal PDF. We show that with larger regularization, GAMMA FFT HA can achieve comparable performance as the best method on the Multi-strategy Counter layout.

> New results for $z$-conditioned policy.


 As requested by reviewer pvTn, we add new results of the $z$-conditioned policy. See Figure (c) and (d) in the rebuttal PDF. In Figure (c), we show the performance of $z$-conditioned Cooperators. In Figure (d), we show the performance of $z$-conditioned decoders. We found that the $z$-conditioned Cooperator can improve performance on the Multi-strategy Counter layout.


> Figure showing humans are adapting during evaluation.


See Fig (e) in the rebuttal PDF.  Human performance improves with the number of trials, indicating that the humans learn, change, and adapt their gameplay during the course of the evaluation. In our evaluation with real humans, each user can change their strategy at any point in the game. A significant proportion of our users self-identify as novice players, both for video games and for Overcooked (17.9% and 49.5%, respectively, Figure f), thus often exhibit improved performance over the course of an increased number of trials. Figure (e) in the rebuttal PDF provides evidence of this pattern, demonstrating that humans change their gameplay style over the course of the evaluation. We hope this help address the concern of reviewer igMC.

> Clarification about how Figure 1 is created.

Reviewer VM8E and UQLZ ask about the details of Figure 1. This is how we get the visualization results in Figure 1.

The point in Figure 1 is the latent vector $z$ of different populations. Taking the population of simulated agents (e.g., the FCP population with 8 FCP agents) as an example, a point is generated by: 1) first, sample an agent in the FCP population; 2) use the agent to generate an episode with self-play; 3) use the VAE encoder to encode the episode into $z$ and map to the 2D space using t-SNE.

For each subfigure, we use the same VAE encoder so all points fall into the same latent space.

---

> ### Comment · Reviewer_igMC · 2024-08-07
> **Concern about statistical significance**
>
> Copying this over from my individual response below:
>
> With regard to standard errors, a gap of one standard error **between the edges of the standard error bars** corresponds to p=0.05. A gap of one standard error **between the means** does **not** correspond to p=0.05. See Fig 5 of the Cumming et al. (2007) paper for an example of this. It is still unclear which results are statistically significant. Consider applying a traditional hypothesis test to get the exact p-values rather than estimating them based on the gap between SEs.

---

> > ### Author Response · Authors · 2024-08-09
> > **Exact p-values.**
> >
> > We appreciate your response to our rebuttal. In order to address your concerns, we provide the exact p-value for each comparison between methods for both layouts. In summary, our method GAMMA is statistically significant (p < 0.05) for the majority of comparisons between methods for both layouts, when comparing based on population generation method, and when comparing our best-performing GAMMA model across all baselines per layout. See the following tables for the exact p-values using a t-test. Our human study data has $n=80$ participants.
> >
> > ### Comparison based on data source:
> > | **Comparison**            | **Counter Circuit** | **Multi Strategy Counter Circuit** |
> > |---------------------------|-------------------------------|----------------------------------------------|
> > | **FCP + GAMMA > FCP**      | **6.48e-5**         | **9.99e-22**       |
> > | **CoMeDi + GAMMA > CoMeDi**| **0.0427**        |  0.902     |
> > | **GAMMA HA DFT/FFT > PPO + BC**| **4.08e-14**    | **0.030**                                            |
> >
> > ### Best-performing GAMMA:
> > | **Comparison**              | **Counter Circuit** | **Multi Strategy Counter Circuit** |
> > |-----------------------------|-------------------------------|----------------------------------------------|
> > | **GAMMA HA DFT/FFT > PPO + BC**  |  **4.08e-14**       | **0.030**   |
> > | **GAMMA HA DFT/FFT > FCP**       | **2.04e-57**    | **3.79e-71**         |
> > | **GAMMA HA DFT/FFT > CoMeDi**    |  **7.14e-07**  | **2.18e-105**       |

---

### Decision · Program_Chairs · 2024-09-25

**Decision:**

Accept (poster)

**Comment:**

The reviews for this paper were split, with two reviewers on the borderline negative side, and two recommending acceptance. The main concerns expressed were insignificant / non-SOTA performance improvement and the 60-second human evaluation setup in Overcooked, which one reviewer argued is too short. (Even if prior work has used 60 seconds, the reviewer argued that the authors should provide their own rationale, which I tend to agree with.) On the other hand, another reviewer strongly championed the paper during the AC-reviewer discussion phase, praising the idea of learning a latent strategy space and arguing that the notion of performance in human-AI collaboration is multifaceted, since explainability, trustworthiness, satisfaction, etc., are all important factors. They also praised the evaluations as being comprehensive relative to other work in this domain. Given the variance in scores, I looked at this paper closely, and broadly agree with the most positive reviewer. This is interesting, mature work and I agree about the comprehensiveness of the evaluation. While the reviewers have pointed out valid minor issues, I don't see any major flaws that warrant blocking publication.